# DROPOUT Q-FUNCTIONS FOR DOUBLY EFFICIENT REINFORCEMENT LEARNING

**Takuya Hiraoka** [1,2], **Takahisa Imagawa** [2], **Taisei Hashimoto** [2,3], **Takashi Onishi** [1,2],
**Yoshimasa Tsuruoka** [2,3]

[1] NEC Corporation
[2] National Institute of Advanced Industrial Science and Technology
[3] The University of Tokyo
{takuya-h1, takashi.onishi}@nec.com, imagawa.t@aist.go.jp,
{hashimoto-taisei388@g.ecc, tsuruoka@logos.t}.u-tokyo.ac.jp

## ABSTRACT

Randomized ensembled double Q-learning (REDQ) (Chen et al., 2021b) has recently achieved state-of-the-art sample efficiency on continuous-action reinforcement learning benchmarks. This superior sample efficiency is made possible by using a large Q-function ensemble. However, REDQ is much less computationally efficient than non-ensemble counterparts such as Soft Actor-Critic (SAC) (Haarnoja et al., 2018a). To make REDQ more computationally efficient, we propose a method of improving computational efficiency called DroQ, which is a variant of REDQ that uses a small ensemble of dropout Q-functions. Our dropout Q-functions are simple Q-functions equipped with dropout connection and layer normalization. Despite its simplicity of implementation, our experimental results indicate that DroQ is doubly (sample and computationally) efficient. It achieved comparable sample efficiency with REDQ, much better computational efficiency than REDQ, and comparable computational efficiency with that of SAC.

## 1 INTRODUCTION

In the reinforcement learning (RL) community, improving **sample efficiency** of RL methods has been important. Traditional RL methods have been shown to be promising for solving complex control tasks such as dexterous in-hand manipulation (OpenAI et al., 2018). However, RL methods generally require millions of training samples to solve a task (Mendonca et al., 2019). This poor sample efficiency of RL methods is a severe obstacle to practical RL applications (e.g., applications on limited computational resources or in real-world environments without simulators). Motivated by these issues, many RL methods have been proposed to achieve higher sample efficiency. For example, Haarnoja et al. (2018a;b) proposed Soft Actor-Critic (SAC), which achieved higher sample efficiency than the previous state-of-the-art RL methods (Lillicrap et al., 2015; Fujimoto et al., 2018; Schulman et al., 2017).

Since 2019, RL methods that use a high **update-to-data (UTD)** ratio to achieve high sample efficiency have emerged. The UTD ratio is the number of updates taken by the agent compared to the number of actual interactions with the environment. A high UTD ratio promotes sufficiently training Q-functions within a few interactions, which leads to sample-efficient learning. Model-Based Policy Optimization (MBPO) (Janner et al., 2019) is a seminal RL method that uses a high UTD ratio of 20–40 and achieves significantly higher sample efficiency than SAC, which uses a UTD ratio of 1. Encouraged by the success of MBPO, many RL methods with high UTD ratios have been proposed (Shen et al., 2020; Lai et al., 2020).

With such methods, randomized ensembled double Q-learning (REDQ) proposed by Chen et al. (2021b) is currently the most sample-efficient method for the MuJoCo benchmark. REDQ uses a high UTD ratio and large ensemble of Q-functions. The use of a high UTD ratio increases an estimation bias in policy evaluation, which degrades sample-efficient learning. REDQ uses an ensemble of Q-functions to suppress the estimation bias and improve sample efficiency. Chen et al. (2021b) demonstrated that the sample efficiency of REDQ is equal to or even better than that of MBPO.

However, REDQ leaves room for improvement in terms of **computational efficiency**. REDQ runs 1.1 to 1.4 times faster than MBPO (Chen et al., 2021b) but is still less computationally efficient than non-ensemble-based RL methods (e.g., SAC) due to the use of large ensembles. In Section 4.2, we show that REDQ requires more than twice as much computation time per update, and much larger memory. Computational efficiency is important in several scenarios, e.g., in RL applications with much lighter on-device computation (e.g. mobile phones or other light-weight edge devices) (Chen et al., 2021a), or in situations in which rapid trials and errors are required for developing RL agents (e.g., hyperparameter tuning or proof of concept for RL applications with limited time resources). Therefore, RL methods that are superior not only in terms of sample efficiency but also in computational efficiency are preferable.

We propose a method of improving computational efficiency. Our method is called DroQ and is a REDQ variant that *uses a small ensemble of dropout Q-functions, in which dropout (Srivastava et al., 2014) and layer normalization (Ba et al., 2016) are used (Section 3).* We experimentally show that DroQ is doubly (computationally and sample) efficient: (i) DroQ significantly improves computational efficiency over REDQ (Section 4.2) by more than two times and (ii) achieves sample efficiency comparable to REDQ (Section 4.1).

Although our primary contribution is **proposing a doubly efficient RL method**, we also make three significant contributions from other perspectives:
**1. Simplicity of implementation.** DroQ can be implemented by basically adding a few lines of readily available functions (dropout and layer normalization) to Q-functions in REDQ (and SAC). This simplicity enables one to easily replicate and extend it.
**2. First successful demonstration of the usefulness of dropout in *high* UTD ratio settings.** Previous studies incorporated dropout into RL (Gal & Ghahramani, 2016; Harrigan, 2016; Moerland et al., 2017; Gal et al., 2016; 2017; Kahn et al., 2017; Jaques et al., 2019; He et al., 2021) (see Section 5 for details). However, these studies focused on *low* UTD ratio settings (i.e., UTD ratio $\leq 1$). Dropout approaches generally do not work as well as ensemble approaches (Osband et al., 2016; Ovadia et al., 2019; Lakshminarayanan et al., 2017; Durasov et al., 2021). For this reason, instead of dropout approaches, ensemble approaches have been used in RL with high UTD ratio settings (Chen et al., 2021b; Janner et al., 2019; Shen et al., 2020; Hiraoka et al., 2021; Lai et al., 2020). In Section 4, we argue that DroQ achieves almost the same or better bias reduction ability and sample/computationally efficiency compared with ensemble-based RL methods in high UTD ratio settings. This sheds light on dropout approaches once again and promotes their use as a reasonable alternative (or complement) to ensemble approaches in high UTD ratio settings.
**3. Discovery of engineering insights to effectively apply dropout to RL.** Specifically, we discovered that the following two engineering practices are effective in reducing bias and improving sample efficiency: (i) using the dropout and ensemble approaches together for constructing Q-functions (i.e., using *multiple* dropout Q-functions) (Section 4.1 and Appendix A.3); and (ii) introducing layer normalization into dropout Q-functions (Section 4.3 and Appendices D, E, and F). These engineering insights were not revealed in previous RL studies and would be useful to practitioners who attempt to apply dropout to RL.

## 2 PRELIMINARIES

### 2.1 MAXIMUM ENTROPY REINFORCEMENT LEARNING (MAXIMUM ENTROPY RL)

RL addresses the problem of an agent learning to act in an environment. At each discrete time step $t$, the environment provides the agent with a state $s_t$, the agent responds by selecting an action $a_t$, and then the environment provides the next reward $r_t$ and state $s_{t+1}$. For convenience, as needed, we use the simpler notations of $r$, $s$, $a$, $s'$, and $a'$ to refer to a reward, state, action, next state, and next action, respectively.

We focus on *maximum entropy* RL, in which an agent aims to find its policy that maximizes the expected return with an entropy bonus: $\arg\max_{\pi} \sum_{t=0}^{T} \gamma^t \mathbb{E}_{s_t,a_t} [r_t + \alpha \mathcal{H}(\pi(\cdot|s_t))]$. Here, $\pi : \mathcal{S} \times \mathcal{A} \to [0, \infty)$ is a policy and $\mathcal{H}$ is entropy. Temperature $\alpha$ balances exploitation and exploration and affects the stochasticity of the policy.

---

**Algorithm 1** REDQ

---

1: Initialize policy parameters $\theta$, $N$ Q-function parameters $\phi_i$, $i = 1, ..., N$, and empty replay buffer $\mathcal{D}$. Set target parameters $\bar{\phi}_i \leftarrow \phi_i$, for $i = 1, ...., N$.
2: **repeat**
3:     Take action $a_t \sim \pi_\theta(\cdot|s_t)$; Observe reward $r_t$, next state $s_{t+1}$; $\mathcal{D} \leftarrow \mathcal{D} \bigcup (s_t, a_t, r_t, s_{t+1})$.
4:     **for** $G$ updates **do**
5:         Sample a mini-batch $\mathcal{B} = \{(s, a, r, s')\}$ from $\mathcal{D}$.
6:         Sample a set $\mathcal{M}$ of $M$ distinct indices from $\{1, 2, ..., N\}$.
7:         Compute the Q target $y$ (same for all $N$ Q-functions):

$$y = r + \gamma \left( \min_{i \in \mathcal{M}} Q_{\bar{\phi}_i}(s', a') - \alpha \log \pi_\theta(a'|s') \right), \ \ a' \sim \pi_\theta(\cdot|s')$$

8:         **for** $i = 1, ..., N$ **do**
9:             Update $\phi_i$ with gradient descent using

$$\nabla_\phi \frac{1}{|\mathcal{B}|} \sum_{(s,a,r,s') \in \mathcal{B}} (Q_{\phi_i}(s, a) - y)^2$$

10:             Update target networks with $\bar{\phi}_i \leftarrow \rho \bar{\phi}_i + (1 - \rho)\phi_i$.
11:     Update $\theta$ with gradient ascent using

$$\nabla_\theta \frac{1}{|\mathcal{B}|} \sum_{s \in \mathcal{B}} \left( \frac{1}{N} \sum_{i=1}^{N} Q_{\phi_i}(s, a) - \alpha \log \pi_\theta(a|s) \right), \ \ a \sim \pi_\theta(\cdot|s)$$

---

## 2.2 RANDOMIZED ENSEMBLED DOUBLE Q-LEARNING (REDQ)

REDQ (Chen et al., 2021b) is a sample-efficient model-free method for solving maximum-entropy RL problems (Algorithm 1). It has two primary components to achieve high sample efficiency.
**1. High UTD ratio:** It uses a high UTD ratio $G$, which is the number of updates (lines 4–10) taken by the agent compared to the number of actual interactions with the environment (line 3). The high UTD ratio promotes sufficient training of Q-functions within a few interactions. However, this also increases an overestimation bias in the Q-function training, which degrades sample-efficient learning (Chen et al., 2021b).
**2. Ensemble of Q-functions:** To reduce the overestimation bias, it uses an ensemble of $N$ Q-functions for the target to be minimized (lines 6–7). Specifically, a random subset $\mathcal{M}$ of the ensemble is selected (line 6) then used for calculating the target (line 7). The size of the subset $\mathcal{M}$ is kept fixed and is denoted as $M$. In addition, each Q-function in the ensemble is randomly and independently initialized but updated with the same target (lines 8–9). Chen et al. (2021b) showed that using a large ensemble ($N = 10$) and small subset ($M = 2$) successfully reduces the bias.

Although using a large ensemble of Q-functions is beneficial for reducing bias and improving sample efficiency, this makes REDQ computationally intensive. In the next section, we discuss reducing the ensemble size.

## 3 INJECTING MODEL UNCERTAINTY INTO TARGET WITH DROPOUT Q-FUNCTIONS

In this section, we discuss replacing the large ensemble of Q-functions in REDQ with a small ensemble of dropout Q-functions. We start our discussion by reviewing what the ensemble in REDQ does from the viewpoint of model uncertainty injection. Then, we propose to use dropout for model uncertainty injection instead of the large ensemble. Specifically, we propose (i) the dropout Q-function that is a Q-function equipped with dropout and layer normalization, and (ii) DroQ, a variant of REDQ that uses a small ensemble of dropout Q-functions. Finally, we explain that the size of the ensemble can be smaller in DroQ than in REDQ.

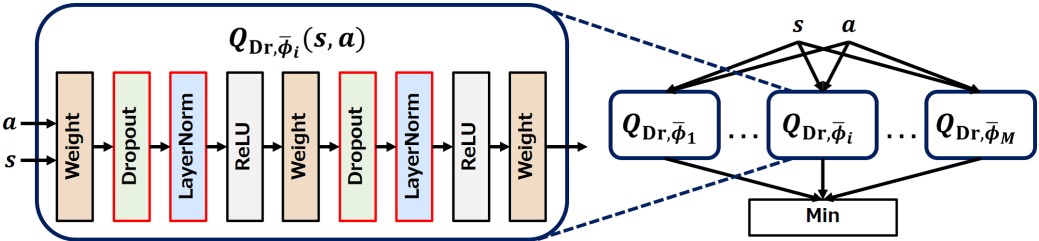

Figure 1: Dropout Q-function implementation (left part) and how dropout Q-functions are used in target (right part). **Dropout Q-function implementation:** Our dropout function is implemented by modifying that used by Chen et al. (2021b). Our modification (highlighted in red) is adding dropout (Dropout) and layer normalization (LayerNorm). "Weight" is a weight layer and "ReLU" is the activation layer of rectified linear units. Parameters $\bar{\phi}_i$ represent the weights and biases in weight layers. **How dropout Q-functions are used in target:** $M$ dropout Q-functions are used to calculate the target value as $\min_{i=1,...,M} Q_{\mathrm{Dr},\bar{\phi}_i}(s, a)$.

We first explain our insight that, in REDQ, model (Q-function parameters') uncertainty is injected into the target. In REDQ, the subset $\mathcal{M}$ of $N$ Q-functions is used to compute the target value $\min_{i\in\mathcal{M}} Q_{\bar{\phi}_i}(s', a')$ (lines 6–7 in Algorithm 1). This can be interpreted as an approximation for $\mathbb{E}_{\bar{\phi}_1,...,\bar{\phi}_M}\left[\min_{i=1,...,M} Q_{\bar{\phi}_i}(s', a')\right]$, the expected target value with respect to model uncertainty:

$$\mathbb{E}_{\bar{\phi}_1,...,\bar{\phi}_M}\left[\min_{i=1,...,M} Q_{\bar{\phi}_i}(s', a')\right] = \int \min_{i=1,...,M} Q_{\bar{\phi}_i}(s', a') \prod_{i=1,...,M} p(\bar{\phi}_i)d\bar{\phi}_i$$

$$\approx \int \min_{i=1,...,M} Q_{\bar{\phi}_i}(s', a') \prod_{i=1,...,M} q_{\mathrm{en}}(\bar{\phi}_i)d\bar{\phi}_i \approx \underbrace{\min_{i\in\mathcal{M}} Q_{\bar{\phi}_i}(s', a')}_{\text{(used in line 7 in Algorithm 1)}}.$$

On the left hand side in the second line, the model distribution $p(\bar{\phi}_i)$ is replaced with a proposal distribution $q_{\mathrm{en}}$, which is based on resampling from the ensemble in line 6 in Algorithm 1[1]. On the right hand side (RHS) in the second line, the expected target value is further approximated by one sample average on the basis of $q_{\mathrm{en}}$. The resulting approximation is used in line 7 in Algorithm 1. In the remainder of this section, we discuss another means to approximate the expected target value.

We use $M$ dropout Q-functions $Q_{\mathrm{Dr}}$ for the target value approximation (Figure 1). Here, $Q_{\mathrm{Dr}}$ is a Q-function that has dropout connections (Srivastava et al., 2014). The left part of Figure 1 shows $Q_{\mathrm{Dr}}$ implementation by adding dropout layers to the Q-function implementation used by Chen et al. (2021b). Layer normalization (Ba et al., 2016) is applied after dropout for more effectively using dropout as with Vaswani et al. (2017); Xu et al. (2019). By using $Q_{\mathrm{Dr}}$, the target value is approximated as

$$\mathbb{E}_{\bar{\phi}_1,...,\bar{\phi}_M}\left[\min_{i=1,...,M} Q_{\bar{\phi}_i}(s', a')\right] \approx \int \min_{i=1,...,M} Q_{\bar{\phi}_i}(s', a') \prod_{i=1,...,M} q_{\mathrm{dr}}(\bar{\phi}_i)d\bar{\phi}_i$$

$$\approx \min_{i=1,...,M} Q_{\mathrm{Dr},\bar{\phi}_i}(s', a').$$

Instead of $q_{\mathrm{en}}$, a proposal distribution $q_{\mathrm{dr}}$ based on the dropout is used on the RHS in the first line. The expected target value is further approximated by one sample average on the basis of $q_{\mathrm{dr}}$ in the second line. We use the resulting approximation $\min_{i=1,...,M} Q_{\mathrm{Dr},\bar{\phi}_i}(s', a')$ for injecting model uncertainty into the target value (the right part of Figure 1). For calculating $\min_{i=1,...,M} Q_{\mathrm{Dr},\bar{\phi}_i}(s', a')$, we use $M$ dropout Q-functions, which have independently initialized and trained parameters $\bar{\phi}_1, ..., \bar{\phi}_M$. Using $M$ dropout Q-functions improves the performance of DroQ (our RL method described in the next paragraph), compared with that using a single dropout Q-function (further details are given in A.3).

We now explain DroQ, in which $Q_{\mathrm{Dr}}$ is used for considering model uncertainty. The algorithmic description of DroQ is shown in Algorithm 2. DroQ is a variant of REDQ, and the modified parts

---

[1] We assume that $\forall_{i=1,...,N}$, $\bar{\phi}_i$ independently follow an identical distribution.

---

**Algorithm 2** DroQ

---

1: Initialize policy parameters $\theta$, $M$ Q-function parameters $\phi_i$, $i = 1, ..., M$, and empty replay buffer $\mathcal{D}$. Set target parameters $\bar{\phi}_i \leftarrow \phi_i$, for $i = 1, ..., M$.
2: **repeat**
3:     Take action $a_t \sim \pi_\theta(\cdot|s_t)$. Observe reward $r_t$, next state $s_{t+1}$; $\mathcal{D} \leftarrow \mathcal{D} \bigcup (s_t, a_t, r_t, s_{t+1})$.
4:     **for** $G$ updates **do**
5:         Sample a mini-batch $\mathcal{B} = \{(s, a, r, s')\}$ from $\mathcal{D}$.
6:         Compute the Q target $y$ for the dropout Q-functions:

$$y = r + \gamma \left( \min_{i=1,...,M} Q_{\mathrm{Dr}, \bar{\phi}_i}(s', a') - \alpha \log \pi_\theta(a'|s') \right), \ \ a' \sim \pi_\theta(\cdot|s')$$

7:         **for** $i = 1, ..., M$ **do**
8:             Update $\phi_i$ with gradient descent using

$$\nabla_\phi \frac{1}{|\mathcal{B}|} \sum_{(s,a,r,s') \in \mathcal{B}} (Q_{\mathrm{Dr}, \phi_i}(s, a) - y)^2$$

9:             Update target networks with $\bar{\phi}_i \leftarrow \rho \bar{\phi}_i + (1 - \rho)\phi_i$.
10:     Update $\theta$ with gradient ascent using

$$\nabla_\theta \frac{1}{|\mathcal{B}|} \sum_{s \in \mathcal{B}} \left( \frac{1}{M} \sum_{i=1}^{M} Q_{\mathrm{Dr}, \phi_i}(s, a) - \alpha \log \pi_\theta(a|s) \right), \ \ a \sim \pi_\theta(\cdot|s)$$

---

from REDQ are highlighted in red in the algorithm. In line 6, $Q_{\mathrm{Dr}}$ is used to inject model uncertainty into the target, as we discussed in the previous paragraph. In lines 8 and 10, $M$ dropout Q-functions are used instead of $N$ Q-functions ($N \geq M$) to make DroQ more computationally efficient.

The ensemble size of the dropout Q-functions for DroQ (i.e., $M$) should be smaller than that of Q-functions for REDQ (i.e., $N$). $M$ for DroQ is equal to the subset size for REDQ, which is not greater than $N$. In practice, $M$ is much smaller than $N$ (e.g., $M = 2$ and $N = 10$ in Chen et al. (2021b)). This reduction in the number of Q-functions makes DroQ more computationally efficient. Specifically, DroQ is faster due to the reduction in the number of Q-function updates (line 8 in Algorithm 2). DroQ also requires less memory for holding Q-function parameters. In Section 4.2, we show that DroQ is computationally faster and less memory intensive than REDQ.

## 4 EXPERIMENTS

We conducted experiments to evaluate and analyse DroQ. In Section 4.1, we evaluate DroQ's performance (sample efficiency and bias-reduction ability). In Section 4.2, we evaluate the computational efficiency of DroQ. In Section 4.3, we present an ablation study for DroQ.

### 4.1 SAMPLE EFFICIENCY AND BIAS-REDUCTION ABILITY OF DROQ

To evaluate the performances of DroQ, we compared DroQ with three baseline methods in MuJoCo benchmark environments (Todorov et al., 2012; Brockman et al., 2016). Following Chen et al. (2021b); Janner et al. (2019), we prepared the following environments: **Hopper**, **Walker2d**, **Ant**, and **Humanoid**. In these environments, we compared the following four methods:
**REDQ:** Baseline method that follows the REDQ algorithm (Chen et al., 2021b) (Algorithm 1).
**SAC:** Baseline method that follows the SAC algorithm (Haarnoja et al., 2018a;b). To improve the sample efficiency of this method, delayed policy update and high UTD ratio $G = 20$ were used, as suggested by Chen et al. (2021b).
**DroQ:** Proposed method that follows Algorithm 2.
**Double uncertainty value network (DUVN):** Baseline method. This method is a DroQ variant that injects model uncertainty into a target in the same manner as the original DUVN (Moerland et al., 2017). The original DUVN uses the output of a single dropout Q-function $Q_{\mathrm{Dr}, \bar{\phi}_1}(s', a')$ without

Table 1: Process times (in msec) per overall loop (e.g., lines 3–10 in Algorithm 2). Process times per Q-functions update (e.g., lines 4–9 in Algorithm 2) are shown in parentheses.

|  | Hopper-v2 | Walker2d-v2 | Ant-v2 | Humanoid-v2 |
|---|---|---|---|---|
| SAC | 910 (870) | 870 (835) | 888 (848) | 893 (854) |
| REDQ | 2328 (2269) | 2339 (2283) | 2336 (2277) | 2400 (2340) |
| DUVN | 664 (636) | 762 (731) | 733 (700) | 692 (660) |
| DroQ | 948 (905) | 933 (892) | 954 (913) | 989 (946) |

layer normalization for target as $y = r + \gamma Q_{\mathrm{Dr}, \bar{\phi}_1}(s', a')$ [2]. We introduce the original DUVN to DroQ by modifying target on line 6 in Algorithm 2 as $y = r + \gamma \left( Q_{\mathrm{Dr}, \bar{\phi}_1}(s', a') - \alpha \log \pi_\theta(a'|s') \right)$. Here, layer normalization is not applied in $Q_{\mathrm{Dr}, \bar{\phi}_1}(s', a')$. We use this method to investigate the performance of the existing *single* dropout Q-function (non-ensemble) method.
Following Chen et al. (2021b), we set the hyperparameters as $G = 20$, $N = 10$, and $M = 2$ for all methods except DUVN ($M = 1$). More detailed hyperparameter settings are given in Appendix H.

The methods were compared on the basis of average return and estimation bias.
**Average return:** An average return over episodes. We regarded 1000 environment steps in Hopper and 3000 environment steps in the other environments as one epoch, respectively. After every epoch, we ran ten test episodes with the current policy and recorded the average return.
**Average/std. bias:** Average and standard deviation of the normalized estimation error (bias) of Q-functions $Q_{\phi_i}$ (Chen et al., 2021b). The error represents how significantly the Q-value estimate differs from the true one. Formally, the error is defined as $|Q^\pi(s, a) - \hat{Q}(s, a)|/\mathbb{E}_{\bar{s}, \bar{a} \sim \pi}[Q^\pi(\bar{s}, \bar{a})]$, where $Q^\pi(s, a)$ is the true Q-value under the current policy $\pi$ and $\hat{Q}(s, a)$ is its estimate. In our experiment, $Q^\pi(s, a)$ was approximated by the discounted Monte Carlo return obtained with $\pi$ in the test episodes. In addition, $\hat{Q}(s, a)$ was evaluated as $\hat{Q}(s, a) = 1/N \sum_{i=1}^N Q_{\phi_i}(s, a)$ for SAC and REDQ and as $\hat{Q}(s, a) = 1/M \sum_{i=1}^M Q_{\mathrm{Dr}, \phi_i}(s, a)$ for DroQ and DUVN, respectively.

The comparison results (Figure 2) indicate that DroQ achieved almost the same level of performance as REDQ. Regarding the average return, REDQ and DroQ achieved almost the same sample efficiency overall. In Walker2d and Ant, their learning curves highly overlapped, and there was no significant difference between them. In Humanoid, REDQ was slightly better than DroQ. In Hopper, DroQ was better than REDQ. In all environments except Hopper, REDQ and DroQ improved their average return significantly earlier than SAC and DUVN. Regarding the bias, REDQ and DroQ consistently kept the value-estimation bias closer to zero than SAC and DUVN in all environments.

## 4.2 COMPUTATIONAL EFFICIENCY OF DROQ

We next evaluated the computational efficiency of DroQ. We compared DroQ with the baseline methods on the basis of the following criteria: (i) **Process time** required for executing methods; (ii) **Number of parameters** of each method; (iii) **Bottleneck memory consumption** suggested from the Pytorch profiler[3]. Bottleneck memory consumption is the maximum memory consumption recorded when running the methods. For evaluation, we ran each method on a machine equipped with two Intel(R) Xeon(R) CPU E5-2667 v4 and one NVIDIA Tesla K80.

Process times per update (numbers in parentheses in Table 1) indicate that DroQ runs more than two times faster than REDQ. DroQ (and SAC) requires process times in the 800–900-msec range. REDQ requires process times in the 2200–2300-msec range. Process times also show that learning Q-functions is dominant in an overall loop. This suggests that using compact (e.g., small numbers of) Q-functions is important for improving overall process times.

The number of parameters and bottleneck memory consumption of each method indicate that DroQ is more memory efficient than REDQ. Regarding the numbers of parameters (Table 2), we can see that those of DroQ (and SAC and DUVN) are about one-fifth those of REDQ. Note that the number of parameters of DroQ is equal to that of SAC since DroQ and SAC use the same number (two) of Q-functions. Regarding the bottleneck memory consumption (Table 3), we can see that that for DroQ (SAC and DUVN) is about one-third that for REDQ. We can also see that the bottleneck

---

[2]Harrigan (2016); He et al. (2021) also propose similar targets.
[3]https://pytorch.org/tutorials/recipes/recipes/profiler_recipe.html

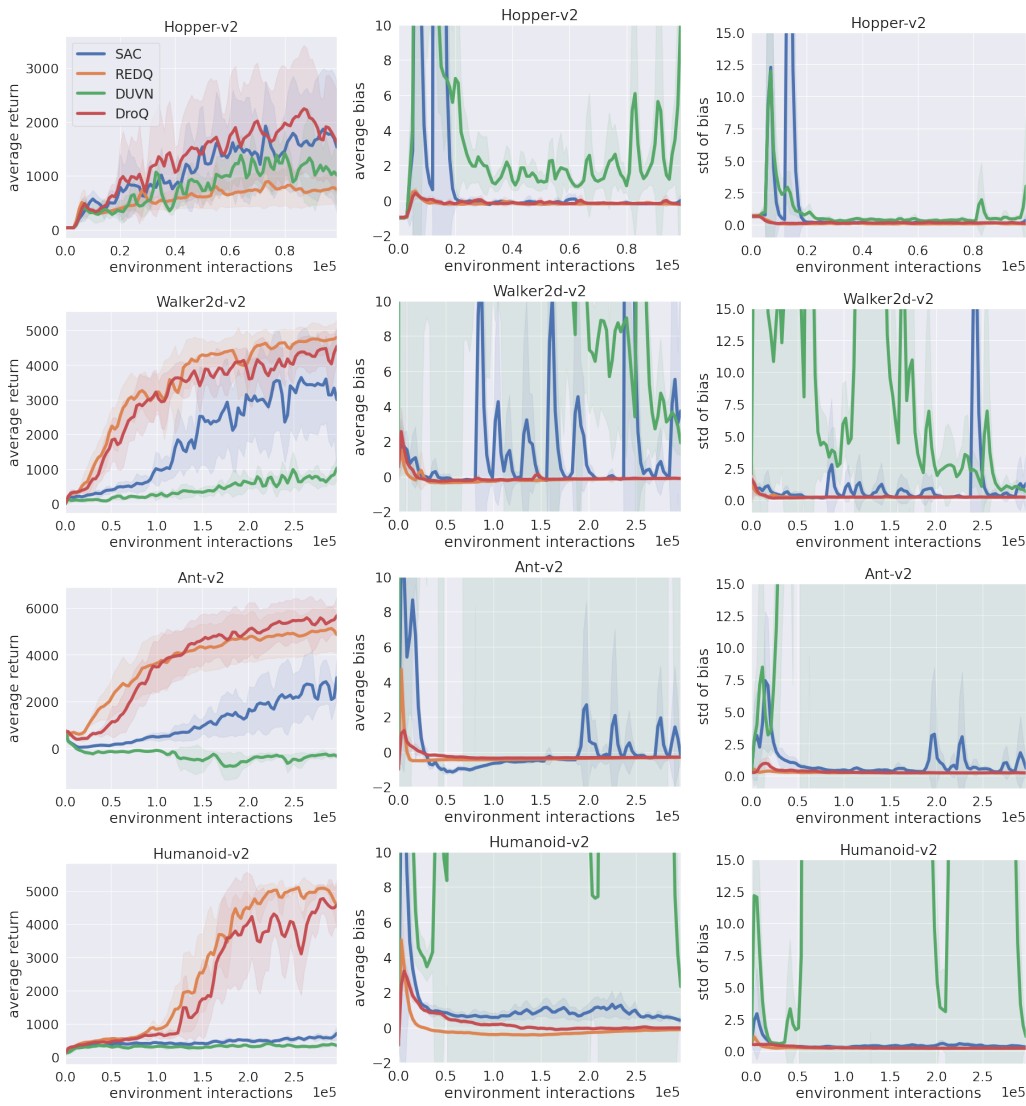

Figure 2: Average return and average/standard deviation of estimation bias for REDQ, SAC, DUVN, and DroQ. The horizontal axis represents the number of interactions with the environment (e.g., the number of executions of line 3 of Algorithm 2). For each method, average score of five independent trials are plotted as solid lines, and standard deviation across trials is plotted as transparent shaded region.

Table 2: Number of parameters

|  | Hopper-v2 | Walker2d-v2 | Ant-v2 | Humanoid-v2 |
|---|---|---|---|---|
| SAC | 141,826 | 146,434 | 152,578 | 166,402 |
| REDQ | 698,890 | 721,930 | 752,650 | 821,770 |
| DUVN | 139,778 | 144,386 | 150,530 | 164,354 |
| DroQ | 141,826 | 146,434 | 152,578 | 166,402 |

memory consumption is almost independent of the environment. This is because one of the most memory-intensive parts is the ReLU activation at the hidden layers in Q-functions[4].

---

[4]In our experiment, the number of units in hidden layer is invariant over environments (Appendix H).

Table 3: Bottleneck memory consumption (in megabytes) with methods. Three worst bottleneck memory consumptions are shown in form of "1st worst bottleneck memory consumption / 2nd worst bottleneck memory consumption / 3rd worst bottleneck memory consumption."

|  | Hopper-v2 | Walker2d-v2 | Ant-v2 | Humanoid-v2 |
|---|---|---|---|---|
| SAC | 73 / 64 / 62 | 73 / 64 / 62 | 73 / 64 / 62 | 73 / 65 / 62 |
| REDQ | 241 / 211 / 200 | 241 / 211 / 200 | 241 / 211 / 200 | 241 / 212 / 201 |
| DUVN | 72 / 71 / 51 | 72 / 71 / 51 | 72 / 71 / 51 | 72 / 71 / 52 |
| DroQ | 73 / 72 / 69 | 73 / 72 / 69 | 73 / 72 / 70 | 73 / 72 / 70 |

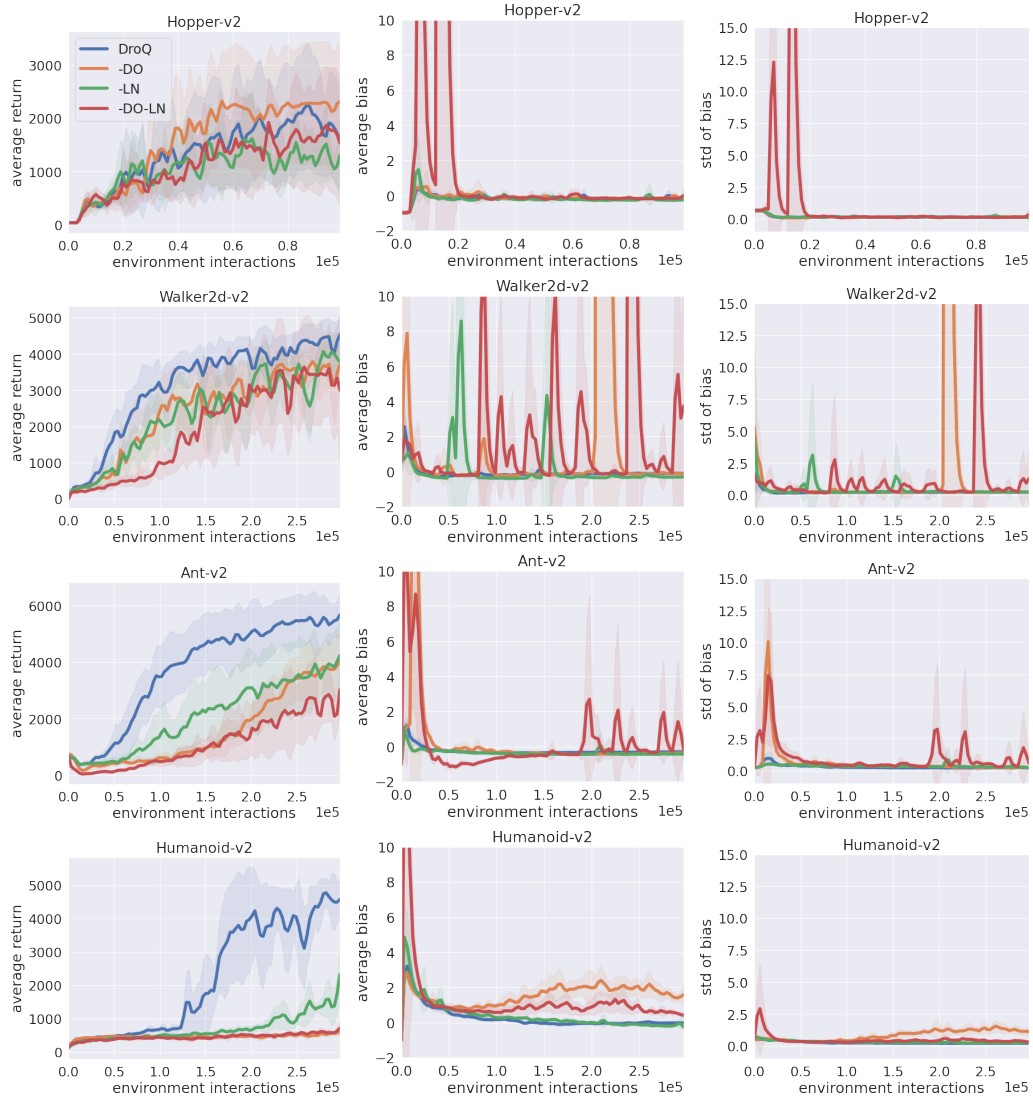

Figure 3: Ablation study results

## 4.3 ABLATION STUDY

As an ablation study of DroQ, we investigated the performance of DroQ variants that do not use either dropout, or layer normalization, or both. We refer to the one without dropout as **-DO**, that without layer normalization as **-LN**, and that without both as **-DO-LN**. The results (Figure 3) indicate that the synergic effect of using dropout and layer normalization is high, especially in complex environments (Ant and Humanoid). In these environments, DroQ significantly outperformed its ablation variants (-DO, -LN and -DO-LN) in terms of both average return and bias reduction.

Table 4: Comparison between related studies and ours. We classify related studies into five types (e.g., "Ensemble Q-functions") on basis of three criteria (e.g. "Type of model uncertainty").

| Type of study | Method for injecting model uncertainty | Type of model uncertainty | Focus on high UTD ratio setting? |
|---|---|---|---|
| Ensemble Q-functions | Ensemble | Q-functions | Partially yes Chen et al. (2021b) |
| Ensemble transition models | Ensemble | Transition models | Partially yes e.g., Janner et al. (2019) |
| Dropout Q-functions | Dropout | Q-functions | No |
| Dropout transition models | Dropout | Transition models | No |
| Normalization | – | – | No |
| Our study | Dropout (with ensemble) | Q-functions | Yes |

## 5  RELATED WORK

In this section, we review related studies and compare them with ours (Table 4).

**Ensemble Q-functions:** Ensembles of Q-functions have been used in RL to consider model uncertainty (Faußer & Schwenker, 2015; Osband et al., 2016; Anschel et al., 2017; Osband et al., 2018; Agarwal et al., 2020; Lee et al., 2021; Lan et al., 2020; Chen et al., 2021b). **Ensemble transition models:** Ensembles of transition (and reward) models have been introduced to model-based RL, e.g., (Chua et al., 2018; Kurutach et al., 2018; Janner et al., 2019; Shen et al., 2020; Yu et al., 2020; Lee et al., 2020; Hiraoka et al., 2021; Abraham et al., 2020). The methods proposed in the above studies use a large ensemble of Q-functions or transition models, thus are computationally intensive. DroQ does not use a large ensemble of Q-functions, and thus is computationally lighter.

**Dropout transition models:** Gal et al. (2016; 2017); Kahn et al. (2017) introduced dropout and its modified variant to transition models of model-based RL methods. **Dropout Q-functions:** Gal & Ghahramani (2016) introduced dropout to a Q-function at action selection for considering model uncertainty in exploration. Harrigan (2016); Moerland et al. (2017); Jaques et al. (2019); He et al. (2021) introduced dropout to policy evaluation in the same vein as us. However, there are three main differences between these studies and ours. (i) Target calculation: they introduced dropout to a *single* Q-function and used it for a target value, whereas we introduce dropout to *multiple* Q-functions and use the minimum of their outputs for the target value. (ii) Use of engineering to stabilize learning: their methods do not use engineering to stabilize the learning of dropout Q-functions, whereas DroQ uses layer normalization to stabilize the learning. (iii) RL setting: they focused on a low UTD ratio setting, whereas we focused on a high UTD ratio setting. A high UTD ratio setting increases a high estimation bias, and thus is a more challenging setting than a low UTD ratio setting. In Sections 4.1 and A.3, we showed that their methods do not perform successfully in a high UTD setting.

**Normalization in RL:** Normalization (e.g., batch normalization (Ioffe & Szegedy, 2015) or layer normalization (Ba et al., 2016)) has been introduced into RL. Batch normalization and its variant are introduced in deep deterministic policy gradient (DDPG) (Lillicrap et al., 2015) and twin delayed DDPG (Fujimoto et al., 2018) (Bhatt et al., 2020). Layer normalization is introduced in the implementation of maximum a posteriori policy optimisation (Abdolmaleki et al., 2018; Hoffman et al., 2020). It is also introduced in SAC extensions (Ma et al., 2020; Zhang et al., 2021). Unlike our study, the above studies did not introduce dropout to consider model uncertainty. In Section 4.3, we showed that using layer normalization alone is not sufficient in high UTD settings. In Appendix E, we also show that using batch normalization does not contribute to performance improvement in high UTD settings.

## 6  CONCLUSION

We proposed, DroQ, an RL method based on a small ensemble of Q-functions that are equipped with dropout connection and layer normalization. We experimentally demonstrated that DroQ significantly improves computational and memory efficiencies over REDQ while achieving sample efficiency comparable with REDQ. In the ablation study, we found that using both dropout and layer normalization had a synergistic effect on improving sample efficiency, especially in complex environments (e.g., Humanoid).

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

# A EFFECT OF DROPOUT RATE ON DROQ AND ITS VARIANTS

## A.1 DROQ

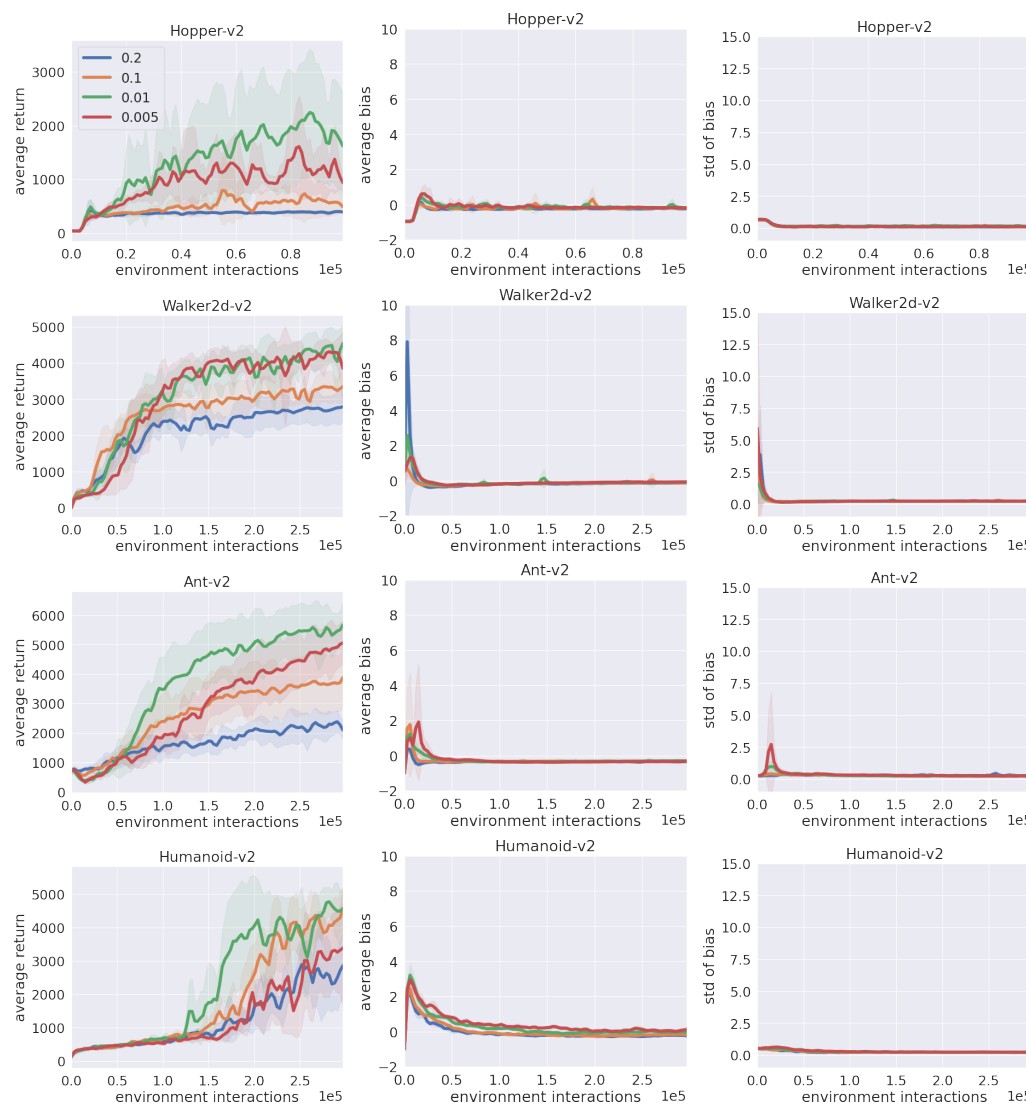

Figure 4: Average return and average/standard deviation of estimation bias for DroQ with different dropout rates. Scores for DroQ are plotted as solid lines and labelled in accordance with dropout rates (e.g.,"0.2").

## A.2 DroQ without layer normalization

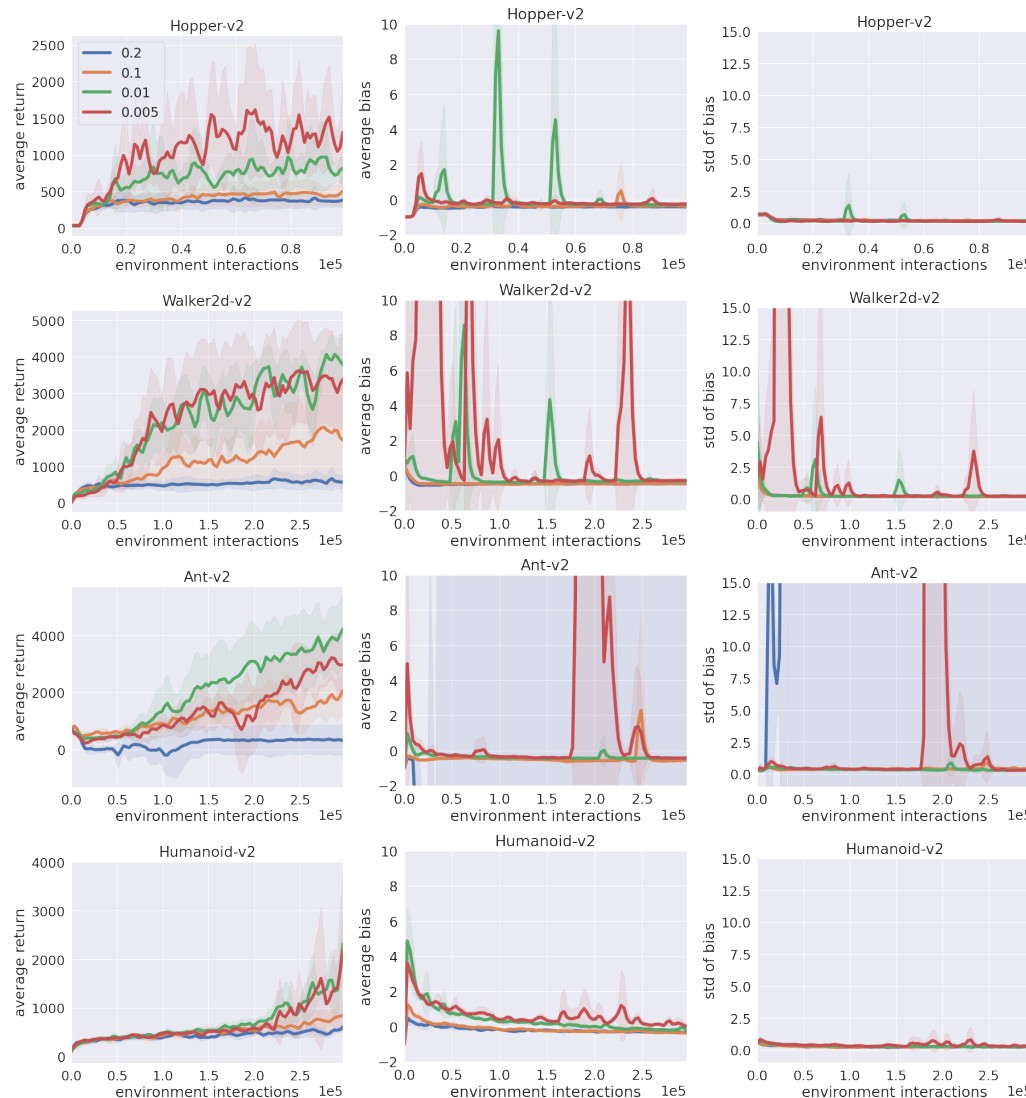

Figure 5: Average return and average/standard deviation of estimation bias for DroQ without layer normalization (i.e., **-LN** in Section 4.3) with different dropout rates

### A.3 SIN-DROQ: DROQ VARIANT USING A SINGLE DROPOUT Q-FUNCTION

DroQ (Algorithm 2) uses an ensemble of multiple ($M$) dropout Q-functions. This raises the question: "Why not use a single dropout Q-function for DroQ?" To answer this question, we compared DroQ with **Sin-DroQ**, a variant of DroQ that uses a single dropout Q-function. Specifically, with Sin-DroQ, the target in line 6 in Algorithm 2 is calculated by evaluating $Q_{\mathrm{Dr}\bar{\phi}_1}(s', a')$, a single dropout Q-function, $M$ times:

$$y = r + \gamma \left( \min(\underbrace{Q_{\mathrm{Dr},\bar{\phi}_1}(s', a'), ..., Q_{\mathrm{Dr},\bar{\phi}_1}(s', a')}_{M \text{ evaluation results of } Q_{\mathrm{Dr},\bar{\phi}_1}(s',a')}) - \alpha \log \pi_\theta(a'|s') \right), \quad a' \sim \pi_\theta(\cdot|s').$$

Note that, the output of $Q_{\mathrm{Dr},\bar{\phi}_1}(s', a')$ can differ in each evaluation due to the use of a dropout connection. It should be also noted that this target is the same as the one proposed in Jaques et al. (2019). The remaining part of Sin-DroQ is the same as DroQ.

From the comparison results of DroQ and Sin-DroQ (Figure 6), we can see that the average return of Sin-DroQ was lower than that of DroQ. This result indicates that using multiple dropout Q-functions is preferable to using a single dropout Q-function with DroQ.

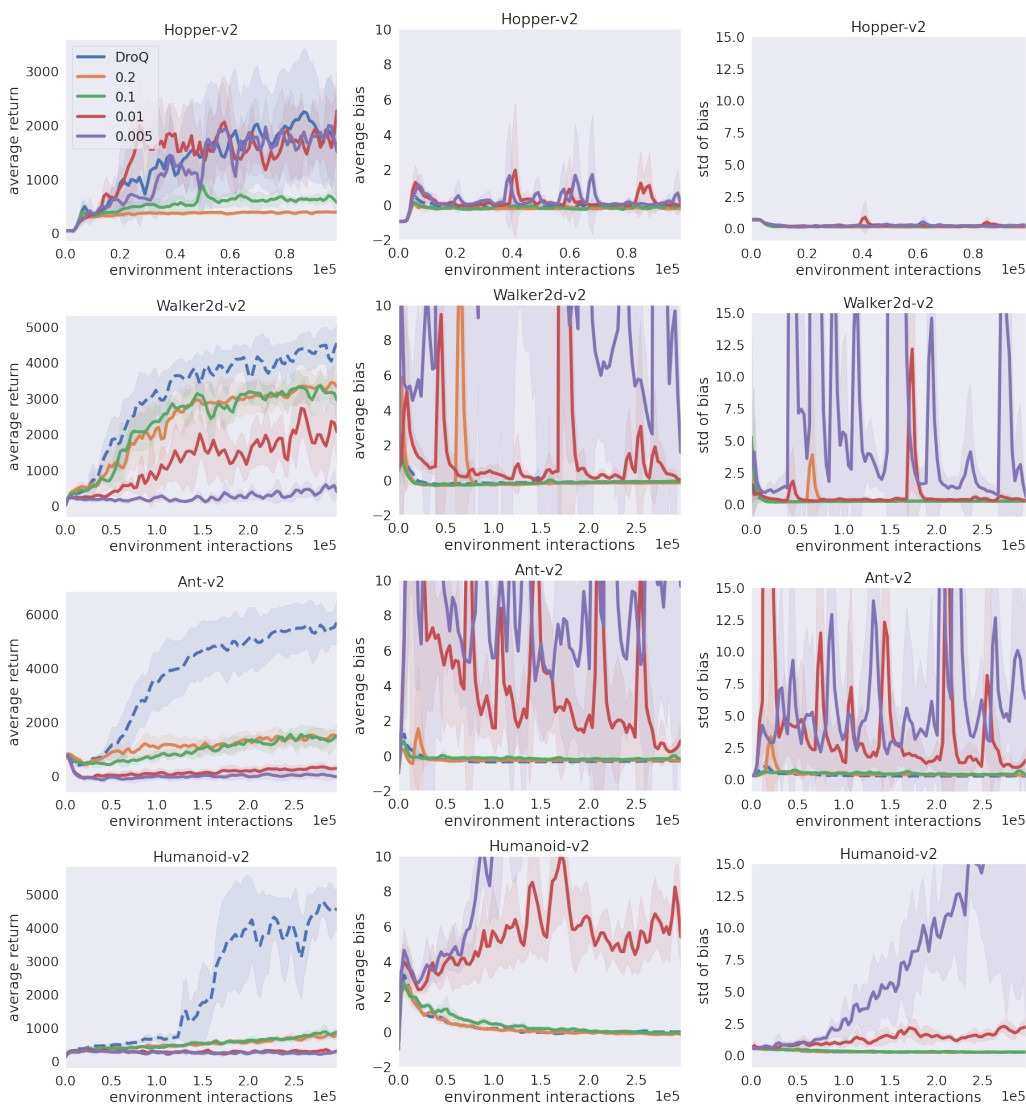

Figure 6: Average return and average/standard deviation of estimation bias for DroQ and Sin-DroQ. Scores for DroQ are plotted as dash lines. Scores for Sin-DroQ are plotted as solid lines and labelled in accordance with dropout rates (e.g., "0.2").

# B   REDQ WITH DIFFERENT ENSEMBLE SIZE $N$

In Section 4, we discussed comparing DroQ with REDQ, which uses an ensemble size of 10 (i.e., $N$=10). To make a more detailed comparison, we compared DroQ and REDQ by varying the ensemble size for REDQ. We denote REDQ that uses an ensemble size of $N$ as "REDQ$N$" (e.g., REDQ5 for REDQ with an ensemble size of five).

Regarding average return (left part of Figure 7), overall, DroQ was superior to REDQ2–5. DroQ was comparable with REDQ2 and REDQ3 in Hopper but superior in more complex environments (Walker, Ant, and Humanoid). In addition, DroQ was somewhat better than REDQ5 in all environments. Regarding estimation bias (middle and right parts of Figure 7), overall, DroQ was significantly better than REDQ2 and comparable with REDQ3–10. Regarding the processing speed (Table 5), DroQ ran as fast as REDQ3 and faster than REDQ5 by 1.4 to 1.5 times. Regarding memory efficiency (Tables 6 and 7), DroQ was less memory intensive than REDQ3–10.

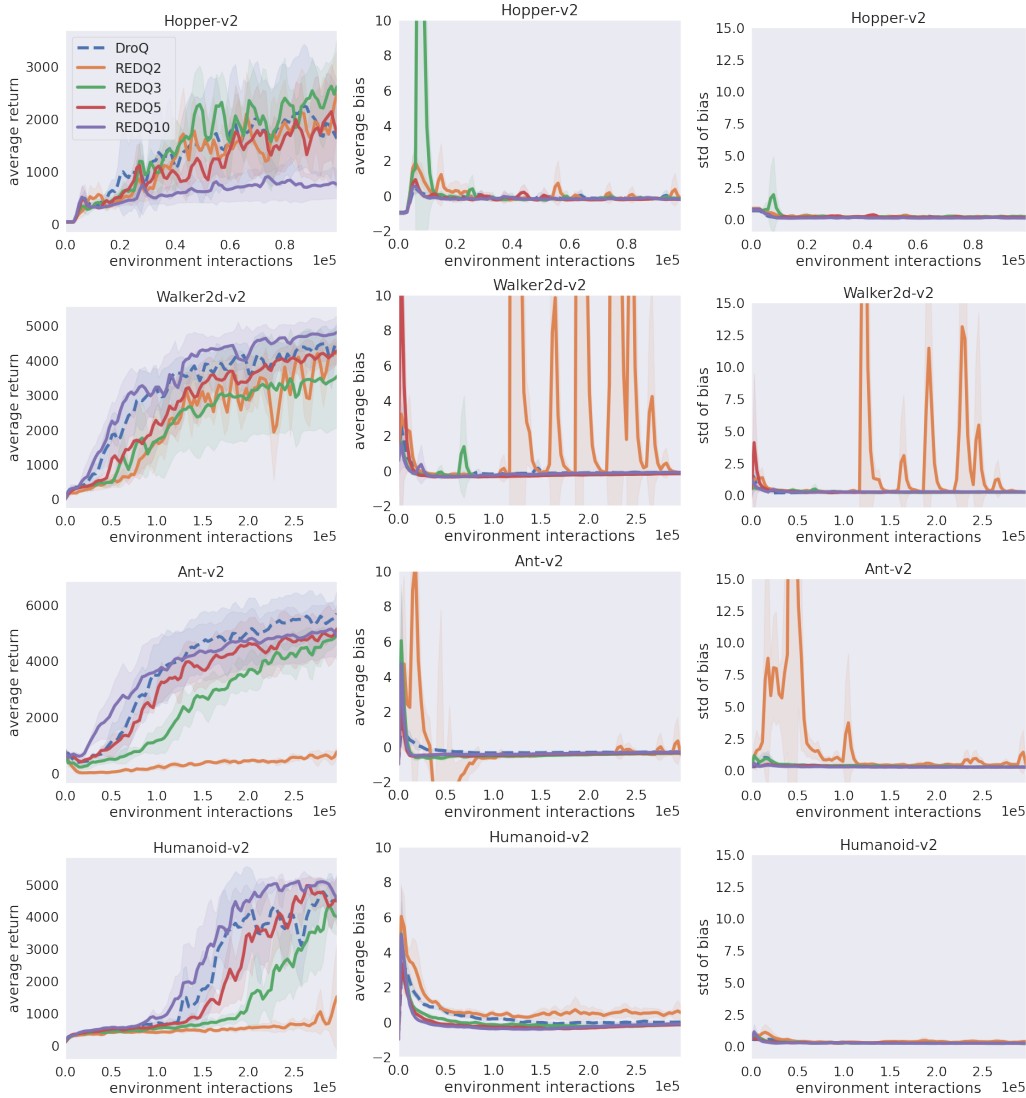

Figure 7: Average return and average/standard deviation of estimation bias for DroQ and REDQ. Scores for DroQ are plotted as dash lines. Scores for REDQ are plotted as solid lines and labelled as "REDQ$N$".

Table 5: Process times (in msec) per overall loop (e.g., lines 3–10 in Algorithm 2). Process times per Q-functions update (e.g., lines 4–9 in Algorithm 2) are shown in parentheses.

|  | Hopper-v2 | Walker2d-v2 | Ant-v2 | Humanoid-v2 |
|---|---|---|---|---|
| DroQ | 948 (905) | 933 (892) | 954 (913) | 989 (946) |
| REDQ2 | 832 (792) | 802 (768) | 675 (641) | 820 (773) |
| REDQ3 | 1052 (1014) | 876 (838) | 919 (881) | 950 (906) |
| REDQ5 | 1414 (1368) | 1425 (1378) | 1373 (1327) | 1552 (1503) |
| REDQ10 | 2328 (2269) | 2339 (2283) | 2336 (2277) | 2400 (2340) |

Table 6: Number of parameters

|  | Hopper-v2 | Walker2d-v2 | Ant-v2 | Humanoid-v2 |
|---|---|---|---|---|
| DroQ | 141,826 | 146,434 | 152,578 | 166,402 |
| REDQ2 | 139,778 | 144,386 | 150,530 | 164,354 |
| REDQ3 | 209,667 | 216,579 | 225,795 | 246,531 |
| REDQ5 | 349,445 | 360,965 | 376,325 | 410,885 |
| REDQ10 | 698,890 | 721,930 | 752,650 | 821,770 |

Table 7: Bottleneck memory consumption (in megabytes) with methods. Three worst bottleneck memory consumptions are shown in form of "1st worst bottleneck memory consumption / 2nd worst bottleneck memory consumption / 3rd worst bottleneck memory consumption."

|  | Hopper-v2 | Walker2d-v2 | Ant-v2 | Humanoid-v2 |
|---|---|---|---|---|
| DroQ | 73 / 72 / 69 | 73 / 72 / 69 | 73 / 72 / 70 | 73 / 72 / 70 |
| REDQ2 | 73 / 51 / 51 | 73 / 51 / 51 | 73 / 51 / 51 | 73 / 52 / 52 |
| REDQ3 | 94 / 64 / 62 | 94 / 64 / 62 | 94 / 64 / 62 | 94 / 65 / 62 |
| REDQ5 | 136 / 106 / 100 | 136 / 106 / 100 | 136 / 106 / 100 | 136 / 107 / 101 |
| REDQ10 | 241 / 211 / 200 | 241 / 211 / 200 | 241 / 211 / 200 | 241 / 212 / 201 |

## C  ADDITIONAL ABLATION STUDY OF DROQ

Dropout is introduced into three parts of the algorithm for DroQ (i.e., lines 6, 8 and 10 of Algorithm 2 in Section 3). In this section, we conducted an ablation study to answer the question "which dropout introduction contributes to overall performance improvement of DroQ?" We compared DroQ with its following variants:

**-DO@TargetQ:** A DroQ variant that does not use dropout in line 6. Specifically, dropout is not used in $Q_{\mathrm{Dr},\bar{\phi}_i}(s',a')$ in the following part in line 6.

$$y = r + \gamma \left( \min_{i=1,\dots,M} Q_{\mathrm{Dr},\bar{\phi}_i}(s',a') - \alpha \log \pi_\theta(a'|s') \right), \ \ a' \sim \pi_\theta(\cdot|s')$$

**-DO@CurrentQ:** A DroQ variant that does not use dropout in line 8. Specifically, dropout is not used in $Q_{\mathrm{Dr},\phi_i}(s,a)$ in the following part in line 8.

$$\nabla_\phi \frac{1}{|\mathcal{B}|} \sum_{(s,a,r,s')\in\mathcal{B}} (Q_{\mathrm{Dr},\phi_i}(s,a) - y)^2$$

**-DO@PolicyOpt:** A DroQ variant that does not use dropout in line 10. Specifically, dropout is not used in $Q_{\mathrm{Dr},\phi_i}(s,a)$ in the following part in line 10.

$$\nabla_\theta \frac{1}{|\mathcal{B}|} \sum_{s\in\mathcal{B}} \left( \frac{1}{M} \sum_{i=1}^{M} Q_{\mathrm{Dr},\phi_i}(s,a) - \alpha \log \pi_\theta(a|s) \right), \ \ a \sim \pi_\theta(\cdot|s)$$

**-DO:** A DroQ variant that does not use dropout in lines 6, 8, and 10.
In this ablation study, we compared the methods on the basis of average return and estimation bias.

The comparison results (Figure 8) indicate that using dropout for both current and target Q-functions (i.e., $Q_{\mathrm{Dr},\bar{\phi}_i}(s',a')$ and $Q_{\mathrm{Dr},\phi_i}(s,a)$ in lines 6 and 8) is effective. Regarding average return, the DroQ variants that do not use dropout for either target Q-functions in line 6 or current Q-functions in line 8 perform significantly worse than that of DroQ. For example, in Ant, the variants that do not use dropout for target Q-functions (-DO@TargetQ and -DO) perform much worse than DroQ. In addition, in Humanoid, the variants that do not use dropout for current Q-functions (-DO@CurrentQ and -DO) perform much worse than DroQ. Regarding estimation bias, that of the variants that do not use dropout for target Q-functions (-DO@TargetQ and -DO) is significantly worse than that of DroQ in all environments.

These results complement the results presented in He et al. (2021). They show that using dropout for both current and target Q-functions improves the performance of the maximum entropy RL method (SAC) in low UTD settings. Our results presented in the previous paragraph implies that their insight (i.e., "using dropout for both current and target Q-functions improves the performance") is valid, even in high UTD settings.

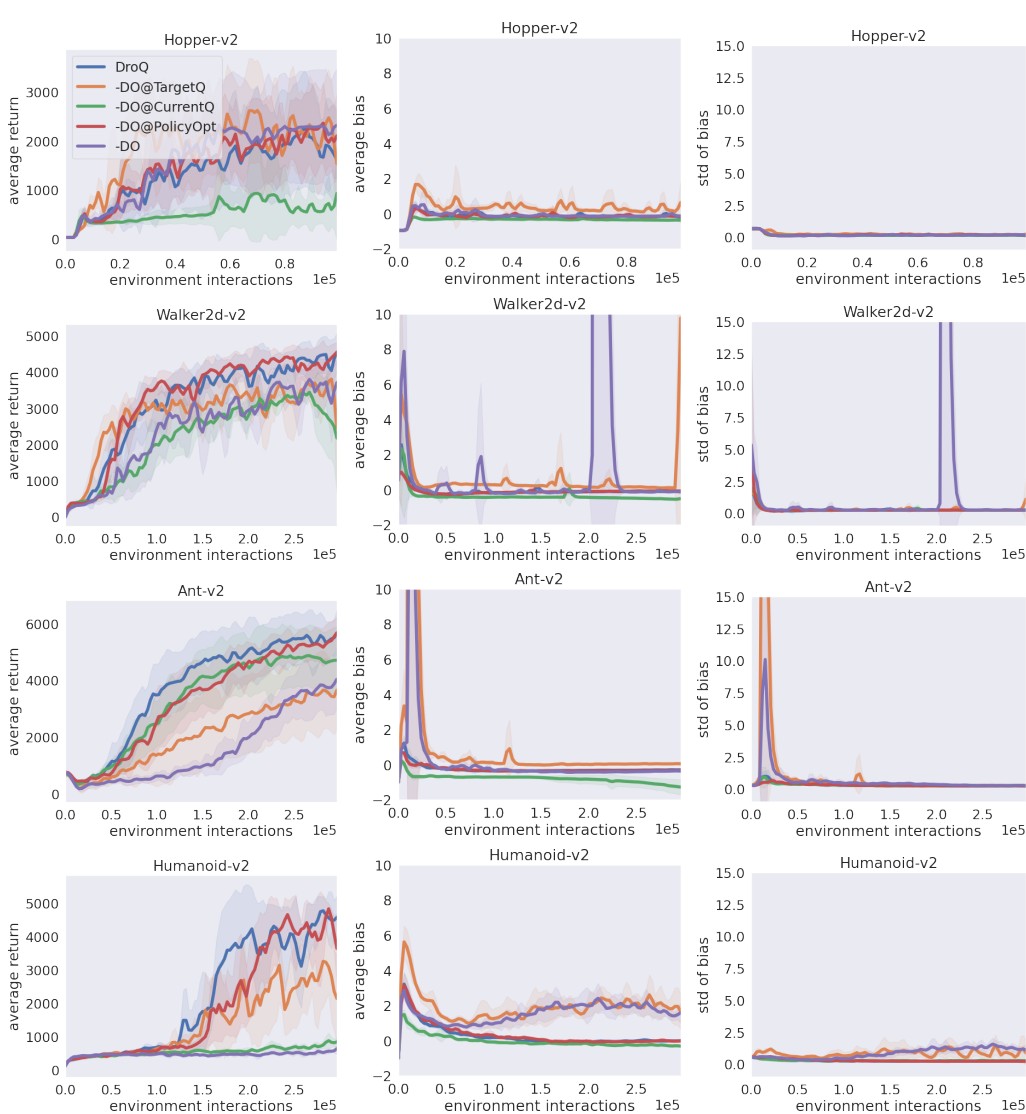

Figure 8: Additional ablation study result about dropout for DroQ

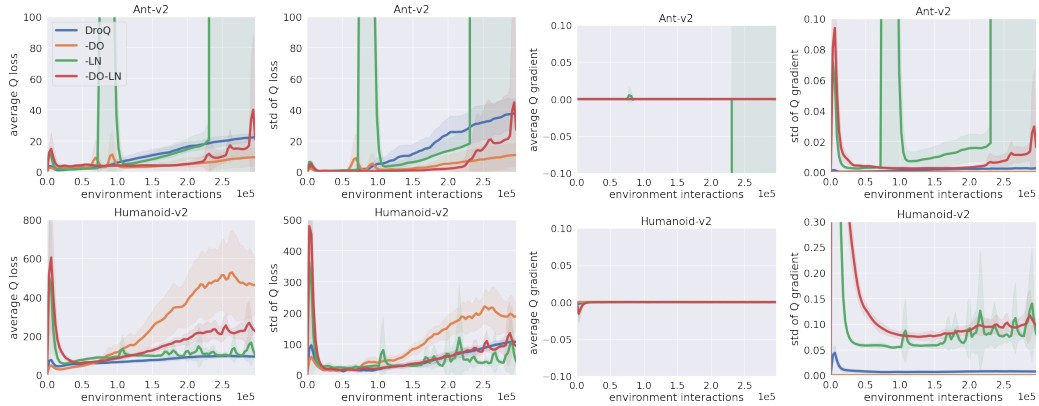

Figure 9: Average/standard deviation of the Q-function loss ("average Q loss" and "std of Q loss") and its gradient with respect to parameters ("average Q gradient" and "std of Q gradient"). Q-function loss is calculated as $\frac{1}{M}\sum_{i=1}^{M} L_i$, where $L_i :=$ $\left(Q_{\phi_i}(s,a) - r - \gamma \left(\min_{i\in\mathcal{M}} Q_{\bar\phi_i}(s',a') - \alpha\log\pi_\theta(a'|s')\right)\right)^2$, and its gradient is calculated as $\frac{1}{M}\sum_{i=1}^{M}\frac{\partial L_i}{\partial\phi_i}$. The definition of method labels (e.g., "DroQ" or "-DO") in the legend is the same as that used in Figure 3.

## D    WHY IS THE COMBINATION OF DROPOUT AND LAYER NORMALIZATION IMPORTANT?

Figure 3 in Section 4.3 showed that combining dropout and layer normalization improves overall performance, especially in complex environments (Ant and Humanoid).

The main reason for this performance improvement could be that layer normalization suppresses the learning instability caused by dropout. Using dropout destabilizes the Q-functions learning. Figure 9 shows that (i) the Q-function loss of the method using dropout (-LN) oscillates more significantly than that of the method not using dropout (-DO-LN), and that (ii) the variance of the gradient loss with respect to the Q-function parameters also oscillates accordingly. Layer normalization suppresses such learning instabilities. The right part of Figure 9 shows that the method using both dropout and layer normalization (DroQ) suppresses the oscillations of the gradient more significantly than the -LN. This stabilization of learning could enable better Q-value propagation and consequently improve overall performance.

# E    EFFECTS OF OTHER NORMALIZATION METHODS

In Section 4.3 and Appendix D, we demonstrated that combining dropout and layer normalization is effective in improving performance. In this section, we examine the effectiveness of normalization methods other than layer normalization.

We examine the effects of (i) batch normalization (Ioffe & Szegedy, 2015), (ii) group normalization (Wu & He, 2018), and (iii) layer normalization without variance re-scaling. Batch normalization is an off-the-shelf and popular normalization method, and has been introduced into RL in low UTD settings (Bhatt et al., 2020). We introduce this popular method to verify whether it is effective in high UTD settings. Group normalization is a method of normalization using the average and variance over the elements of a target input, which is similar to layer normalization[5]. We introduce this method to verify whether a method equipped with a similar mechanism to layer normalization works well in high UTD settings. Layer normalization without variance re-scaling is the layer normalization variant that does not re-scale the target input's variance. Xu et al. (2019) pointed out that variance re-scaling is an important component for layer normalization to work well. We introduce this method to verify whether variance re-scaling is an important component also in our settings.

We compare six DroQ variants that use these three normalization methods. We denote -DO[6] variants using batch normalization, group normalization, and layer normalization without variance re-scaling, instead of layer normalization, as +BN, +GN, and +LNwoVR, respectively. We also denote DroQ variants using batch normalization, group normalization, and layer normalization without variance re-scaling, instead of layer normalization, as +DO+BN, +DO+GN, and +DO+LNwoVR, respectively. We compare these six methods (+BN, +GN, +LNwoVR, +DO+BN, +DO+GN, and +DO+LNwoVR).

From the experimental results of the six methods (Figures 10 and 11), we get the following insights:
1. Batch normalization does not work well. Figure 10 shows that the methods using batch normalization (+BN and +DO+BN) do not significantly improve the average return and estimation bias in Humanoid and Ant. Further, Figure 11 shows that the Q-function learning with these methods is very unstable.
2. Group normalization, which has a similar mechanism to layer normalization, works well when combined with dropout. Figure 10 shows that the method using the group normalization and dropout (+DO+GN) successfully improves the average return and estimation bias in Humanoid and Ant. The figure also shows that the method that uses only group normalization (+GN) fails in improving the average return and estimation bias.
3. Layer normalization without variance re-scaling does not work well. Figure 10 shows that the methods that introduce this layer normalization variant (+LNwoVR and +DO+LNwoVR) do not significantly improve the average return and estimation bias well especially in Humanoid. Further, Figure 11 shows that the Q-function learning is unstable compared to +DO+GN (and DroQ in Figure 9) in both Ant and Humanoid. This result implies that one of the primal factors for the synergistic effect of dropout and layer normalization is the variance re-scaling.

---

[5]Layer normalization normalizes all input elements using their mean and variance. On the other hand, group normalization divides the input elements into subgroups (two groups in our setting) and normalizes each group using the intra-group mean and variance.

[6]DroQ without dropout, which is introduced in Sections 4.3 and D.

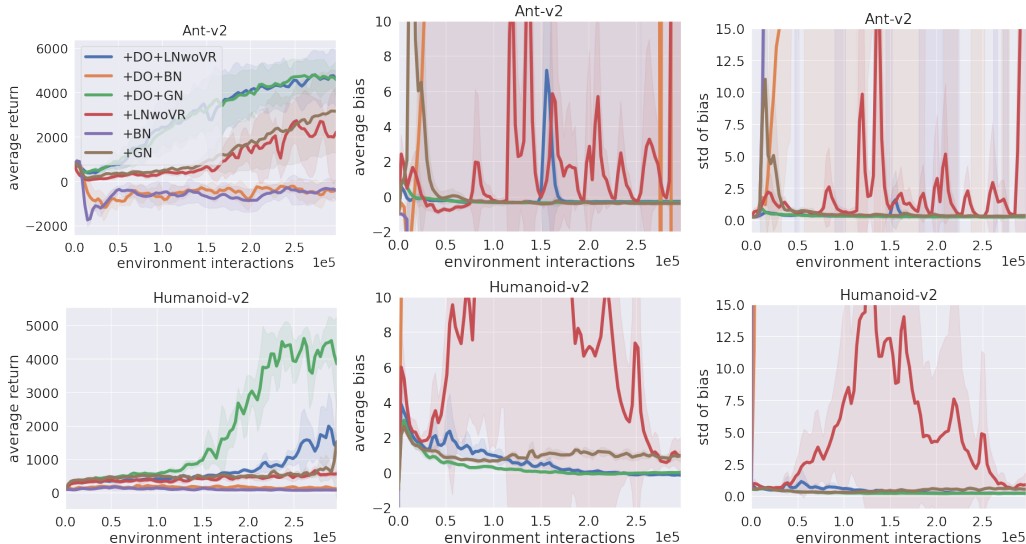

Figure 10: Average return and average/standard deviation of estimation bias for six methods using batch normalization, group normalization, and the variant of layer normalization.

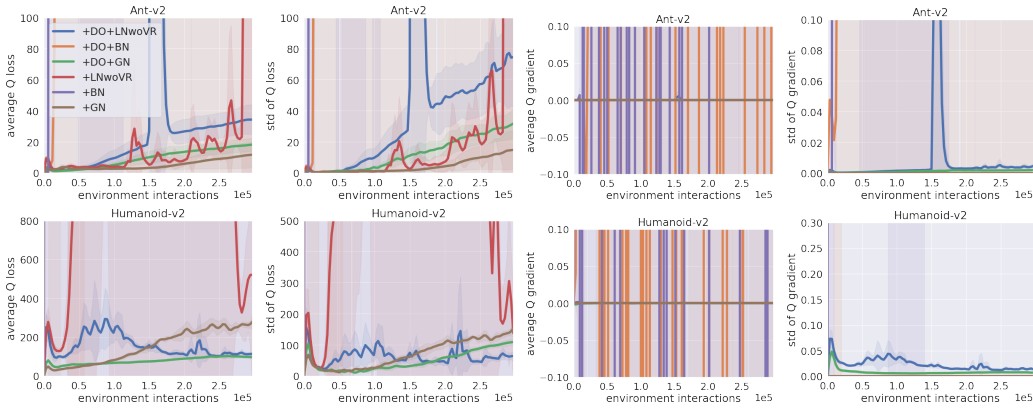

Figure 11: Average/variance of the Q-function loss (left part) and its gradient with respect to parameters (right part).

---

**Algorithm 3** DroQ with ensemble of $N$ dropout Q-functions (DroQ$N$)

---

1: Initialize policy parameters $\theta$, $N$ Q-function parameters $\phi_i$, $i = 1, ..., N$, and empty replay buffer $\mathcal{D}$. Set target parameters $\bar{\phi}_i \leftarrow \phi_i$, for $i = 1, ..., N$.
2: **repeat**
3:     Take action $a_t \sim \pi_\theta(\cdot|s_t)$. Observe reward $r_t$, next state $s_{t+1}$; $\mathcal{D} \leftarrow \mathcal{D} \bigcup (s_t, a_t, r_t, s_{t+1})$.
4:     **for** $G$ updates **do**
5:         Sample a mini-batch $\mathcal{B} = \{(s, a, r, s')\}$ from $\mathcal{D}$.
6:         Sample a set $\mathcal{M}$ of $M$ distinct indices from $\{1, 2, ..., N\}$.
7:         Compute the Q target $y$ for the dropout Q-functions:

$$y = r + \gamma \left( \min_{i \in \mathcal{M}} Q_{\text{Dr}, \bar{\phi}_i}(s', a') - \alpha \log \pi_\theta(a'|s') \right), \quad a' \sim \pi_\theta(\cdot|s')$$

8:         **for** $i = 1, ..., N$ **do**
9:             Update $\phi_i$ with gradient descent using

$$\nabla_\phi \frac{1}{|\mathcal{B}|} \sum_{(s,a,r,s') \in \mathcal{B}} (Q_{\text{Dr}, \phi_i}(s, a) - y)^2$$

10:           Update target networks with $\bar{\phi}_i \leftarrow \rho \bar{\phi}_i + (1 - \rho) \phi_i$.
11:     Update $\theta$ with gradient ascent using

$$\nabla_\theta \frac{1}{|\mathcal{B}|} \sum_{s \in \mathcal{B}} \left( \frac{1}{N} \sum_{i=1}^{N} Q_{\text{Dr}, \phi_i}(s, a) - \alpha \log \pi_\theta(a|s) \right), \quad a \sim \pi_\theta(\cdot|s)$$

---

# F    RELATION BETWEEN (1) ENSEMBLE SIZE AND (2) THE EFFECT OF LAYER NORMALIZATION AND DROPOUT ON OVERALL PERFORMANCE

In Section 4.3 and Appendix D, we showed that using layer normalization and dropout improves overall performance. Further, in Sections 4.1 and A.3, we showed that using multiple (two) dropout Q-functions improves overall performance. These results raise two new questions: (i) *what happens if we use more than three dropout Q-functions?* and (ii) *what the effect of layer normalization and dropout would be in such cases?* In this section, we conduct an additional experiment to answer questions (i) and (ii).

For our additional experiment, we introduce the **DroQ with ensemble of $N$ dropout Q-functions** (DroQ$N$) algorithm (Algorithm 3). This algorithm is a variant of DroQ algorithm (Algorithm 2) which uses bootstrapping of a subset $\mathcal{M}$ of $N$ dropout Q-functions (line 6). This algorithm can also be regarded as a variant of REDQ algorithm (Algorithm 1) which uses $N$ dropout Q-functions. With DroQ$N$ algorithm, the ensemble size can be changed by varying the value of $N$.

We compare the method based on the **DroQ$N$** algorithm and its ablation variants:
**DroQ$N$-LN:** the DroQ$N$ variant that does not use layer normalization for the dropout Q-function $Q_{\text{Dr}, \phi_i}$.
**DroQ$N$-DO-LN:** the DroQ$N$ variant that does not use layer normalization and dropout for $Q_{\text{Dr}, \phi_i}$. Note that DroQ$N$-DO-LN is identical with REDQ$N$ presented in Appendix B.
We evaluate the performance of these methods for the case of $N = 2, 3, 5, 10$.

Learning curves of the methods are shown in Figure 12.
**For the question (i),** in complex environments (Ant and Humanoid), increasing the size of the ensemble improves the sample efficiency. In Ant, DroQ5–10 are more sample efficient than DroQ2–3. In Humanoid, the sample efficiency is monotonically improved when the ensemble size of DroQ$N$ is increased as $N = 2, 3, 5, 10$.
**For the question (ii),** in the complex environments, the synergy of layer normalization and dropout is significant especially when the value of $N$ is small ($N = 2, 3$). In Ant and Humanoid, DroQ2–3 significantly improves the sample efficiency compared to DroQ2–3-DO. On the other hand, the effect of layer normalization itself becomes dominant when the value of $N$ is large ($N \geq 5$). For

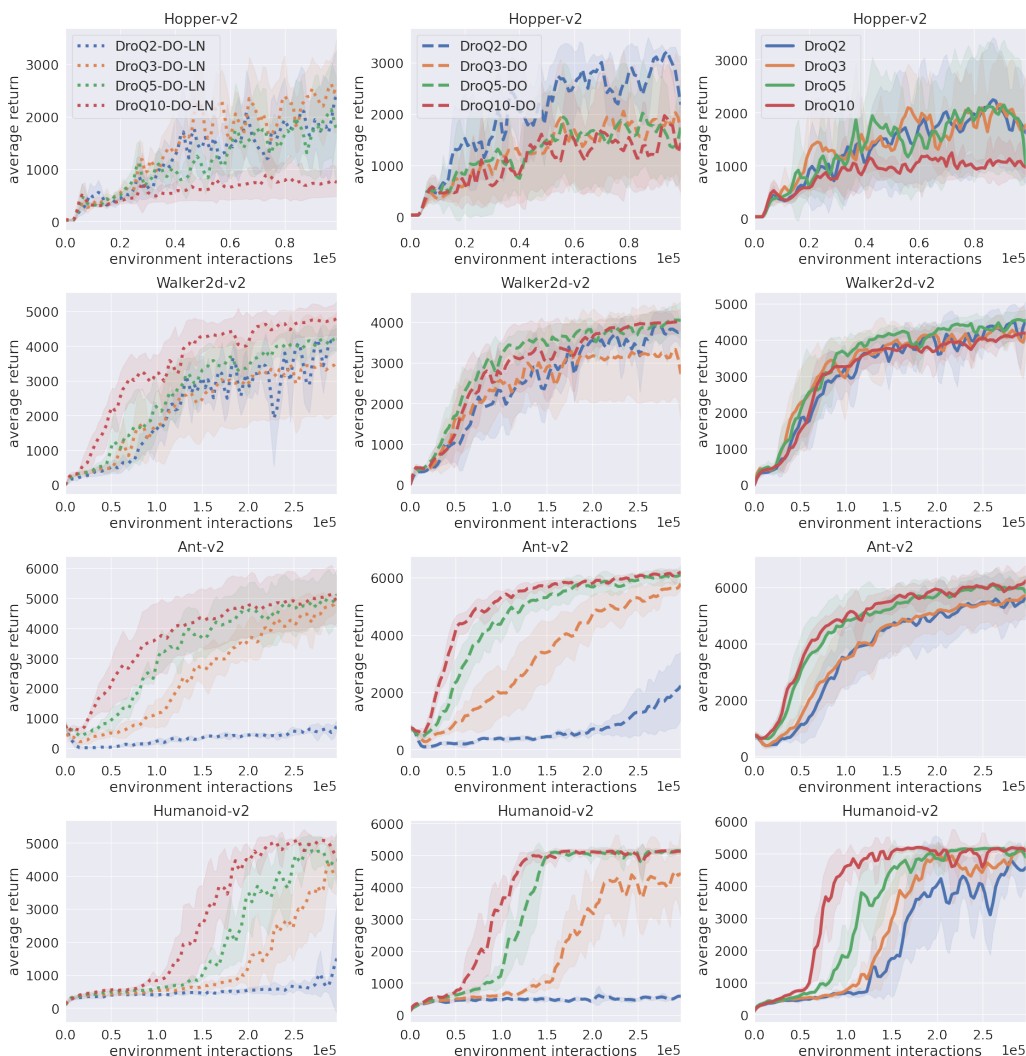

Figure 12: Additional ablation study result (average return). Left part (dotted lines): DroQ$N$-DO-LN, middle part (dashed lines): DroQ$N$-LN, right part (solid lines): DroQ$N$

Ant and Humanoid, we can see that DroQ5–10-DO significantly improves the sample efficiency compared to DroQ5–10-DO-LN. However, DroQ5–10 does not significantly improve the sample efficiency compared to DroQ5–10-DO.

Based on the above result, we suggest (1) using layer normalization and dropout together for small ensemble cases, and (2) using layer normalization alone for large ensemble cases.

## G    EFFECT OF DROPOUT Q-FUNCTIONS ON SAC

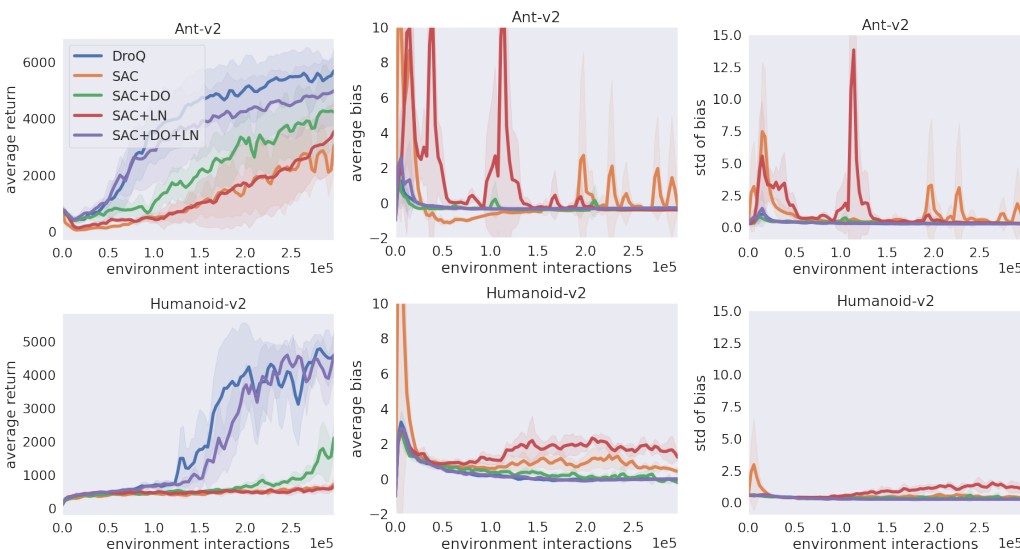

Figure 13: Average return and average/standard deviation of estimation bias for DroQ, SAC, and its variants. SAC+DO is a SAC variant using dropout, SAC+LN is a SAC variant using layer normalization, and SAC+DO+LN is a SAC variant using dropout and layer normalization. Note that the only difference between DroQ ($M = 2$) and SAC+DO+LN is the policy update part (line 10 in Algorithm 2). DroQ ($M = 2$) updates policy as $\nabla_\theta \frac{1}{|B|} \sum_{s \in \mathcal{B}} \left( \frac{1}{2} \sum_{i=1}^{2} Q_{\mathrm{Dr},\phi_i}(s,a) - \alpha \log \pi_\theta(a|s) \right)$. On the other hand, SAC+DO+LN updates policy as $\nabla_\theta \frac{1}{|B|} \sum_{s \in \mathcal{B}} \left( \frac{1}{2} \min_{i=1}^{2} Q_{\mathrm{Dr},\phi_i}(s,a) - \alpha \log \pi_\theta(a|s) \right)$.

# H   HYPERPARAMETER SETTINGS

The hyperparameter settings for each method in the experiments discussed in Section 4 are listed in Table 8. Parameter values, except for (i) dropout rate for DroQ and DUVN and (ii) $M$ for DUVN, were set according to Chen et al. (2021b). The dropout rate (i) was set through line search, and $M$ for DUVN (ii) was set according to Harrigan (2016); Moerland et al. (2017).

Table 8: Hyperparameter settings

| Method | Parameter | Value |
|---|---|---|
| SAC, REDQ, DroQ, and DUVN | optimizer | Adam (Kingma & Ba, 2015) |
| | learning rate | $3 \cdot 10^2$ |
| | discount rate ($\gamma$) | 0.99 |
| | target-smoothing coefficient ($\rho$) | 0.005 |
| | replay buffer size | $10^6$ |
| | number of hidden layers for all networks | 2 |
| | number of hidden units per layer | 256 |
| | mini-batch size | 256 |
| | random starting data | 5000 |
| | UTD ratio $G$ | 20 |
| REDQ and DroQ | in-target minimization parameter $M$ | 2 |
| REDQ | ensemble size $N$ | 10 |
| DroQ and DUVN | dropout rate | 0.01 |
| DUVN | in-target minimization parameter $M$ | 1 |

# I   EXPERIMENTS ON THE ORIGINAL REDQ CODEBASE

Experiments presented in the main content of this paper have been implemented on top of the SAC codebase (`https://github.com/ku2482/soft-actor-critic.pytorch`). In this section, we report the results of our experiments on the original REDQ codebase (`https://github.com/watchernyu/REDQ`). Our experiments are to replicate experiments in the main content of this paper. To evaluate the computational efficiency, we plot the wallclock time required to complete a certain number of interactions with the environment, instead of the standard deviation of bias. In addition, dropout rate values are re-tuned for each environment (Table 9). Figure 14 shows the replica of the results presented in Section 4.1. Figure 15 shows the replica of the results provided in Section 4.3. Figure 16 shows the replica of the results provided in Section A.1. Figure 17 shows the replica of the results provided in Section D. Figures 18 and 19 show the replica of the results provided in Section E. Figure 20 shows the replica of the results provided in Section F.

Table 9: Dropout rate setting for Appendix I

| Environment | Value |
|---|---|
| Hopper-v2 | 0.0001 |
| Walker2d-v2 | 0.005 |
| Ant-v2 | 0.01 |
| Humanoid-v2 | 0.1 |

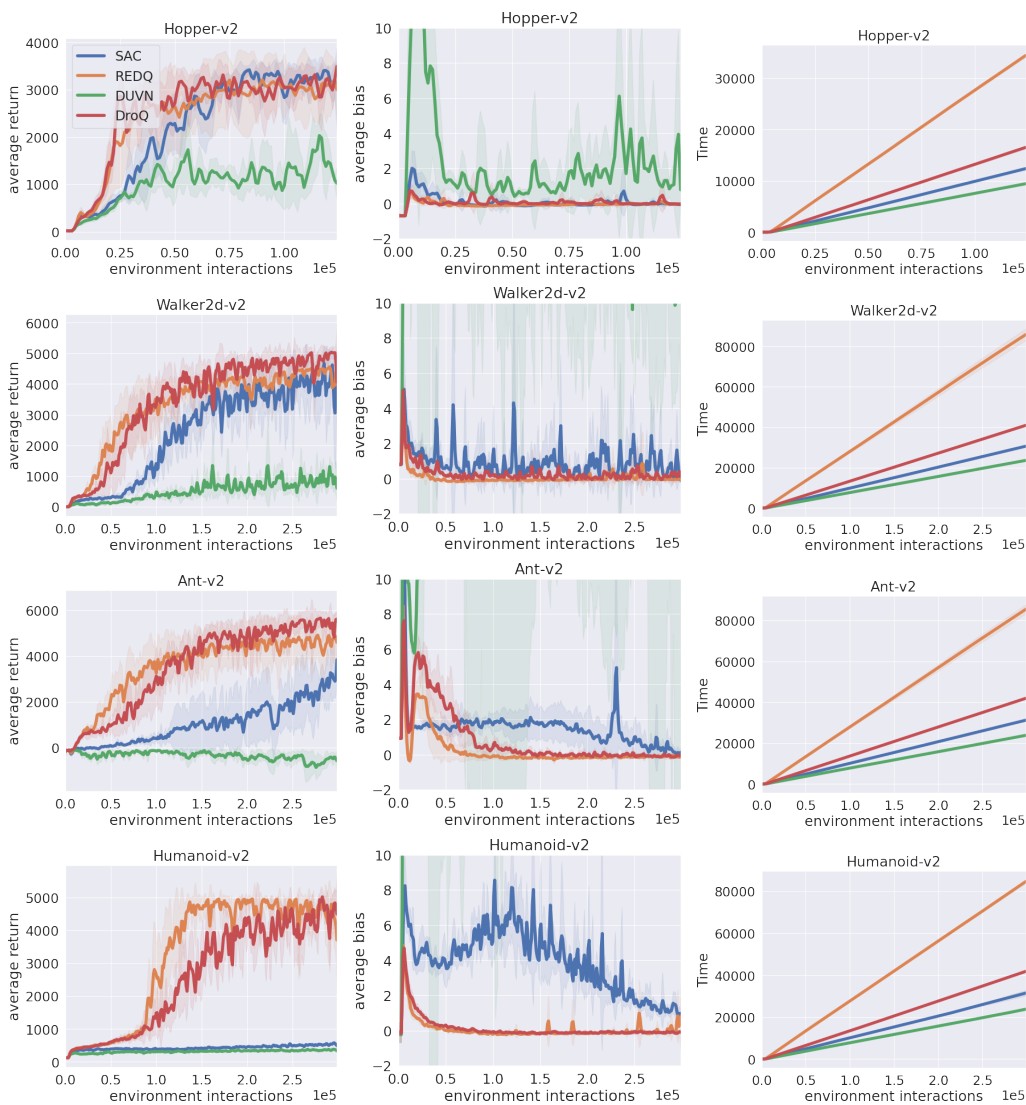

Figure 14: Average return, average of estimation bias, and wallclock time (second) for REDQ, SAC, DUVN, and DroQ. The horizontal axis represents the number of interactions with the environment (e.g., the number of executions of line 3 of Algorithm 2). For each method, average score of five independent trials are plotted as solid lines, and standard deviation across trials is plotted as transparent shaded region.

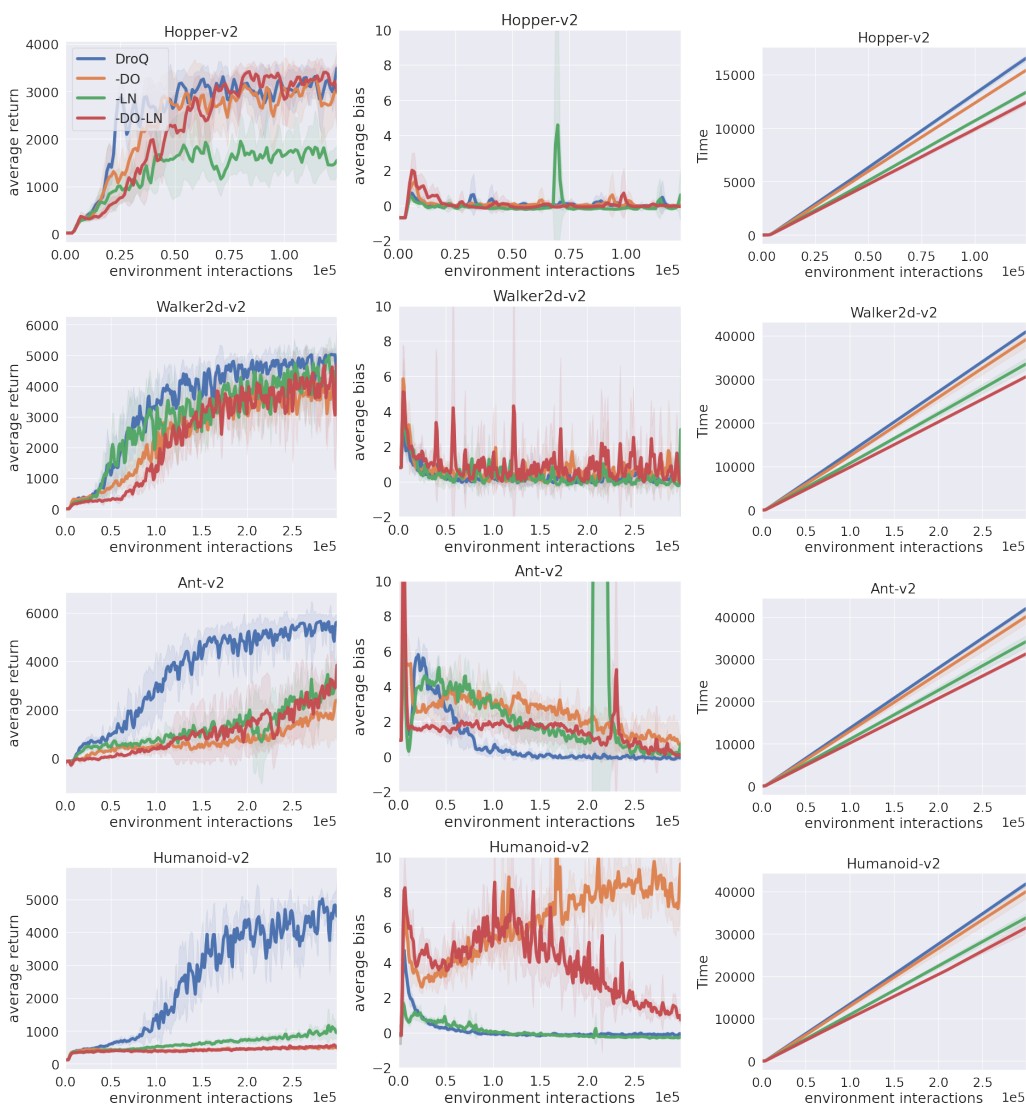

Figure 15: Ablation study results

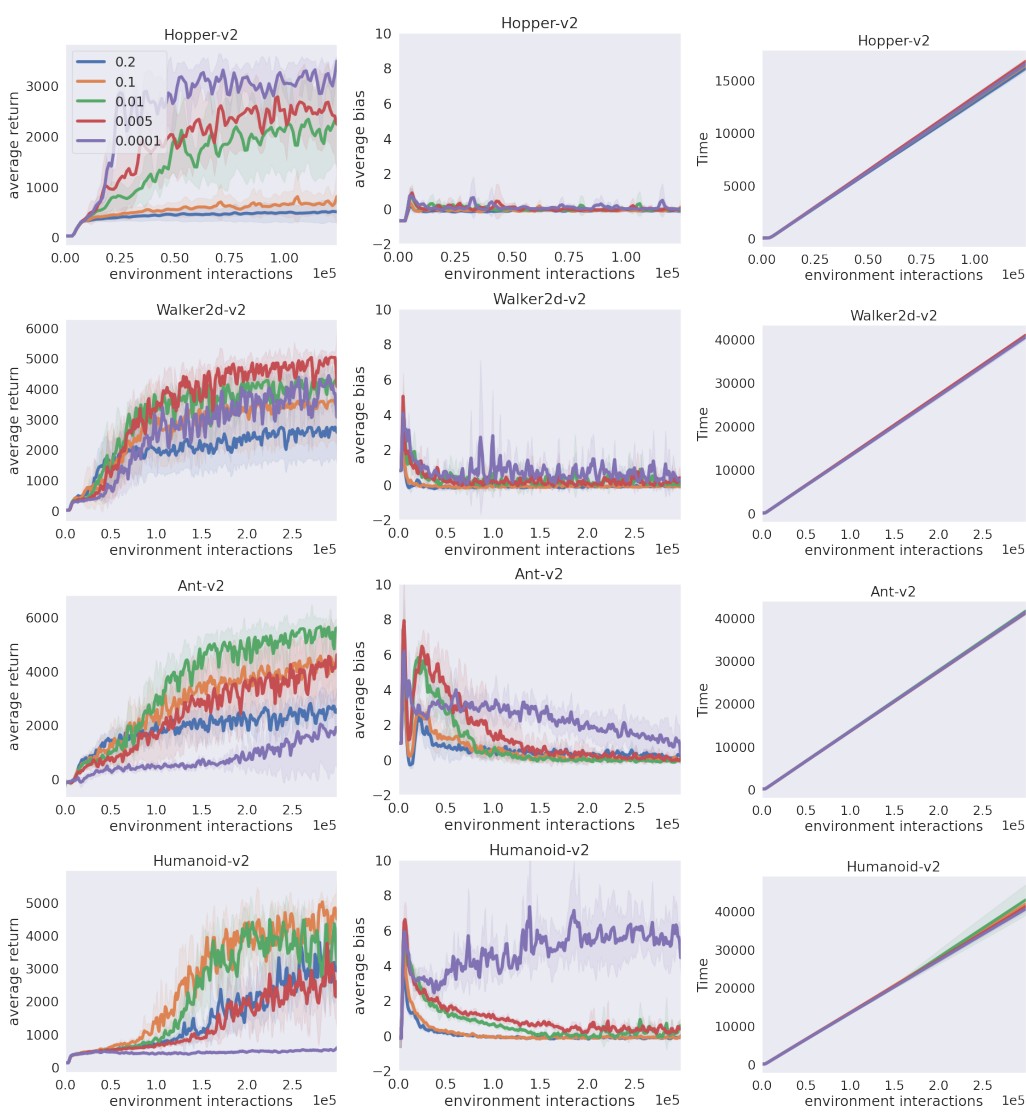

Figure 16: Average return, average of estimation bias, and wallclock time (second) for DroQ with different dropout rates. Scores for DroQ are plotted as solid lines and labelled in accordance with dropout rates (e.g.,"0.2").

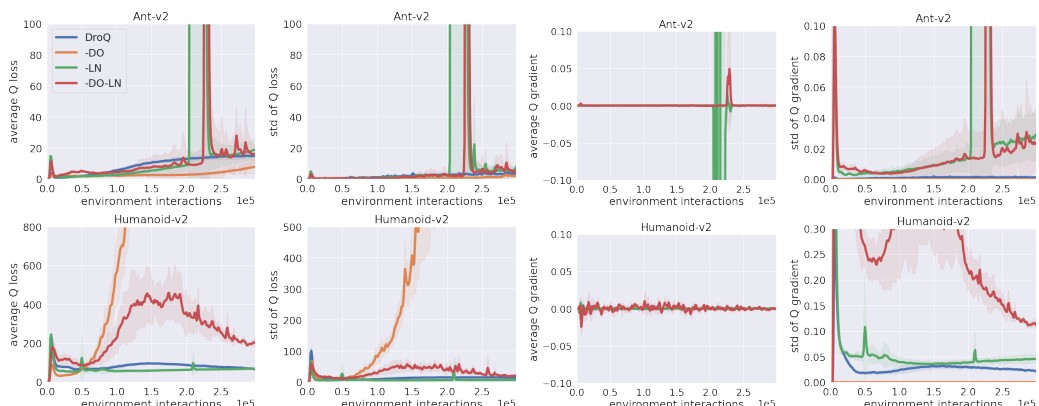

Figure 17: Average/standard deviation of the Q-function loss ("average Q loss" and "std of Q loss") and its gradient with respect to parameters ("average Q gradient" and "std of Q gradient"). Q-function loss is calculated as $\frac{1}{M}\sum_{i=1}^{M} L_i$, where $L_i :=$ $\left(Q_{\phi_i}(s,a) - r - \gamma\left(\min_{i\in\mathcal{M}} Q_{\bar{\phi}_i}(s',a') - \alpha\log\pi_\theta(a'|s')\right)\right)^2$, and its gradient is calculated as $\frac{1}{M}\sum_{i=1}^{M}\frac{\partial L_i}{\partial \phi_i}$.

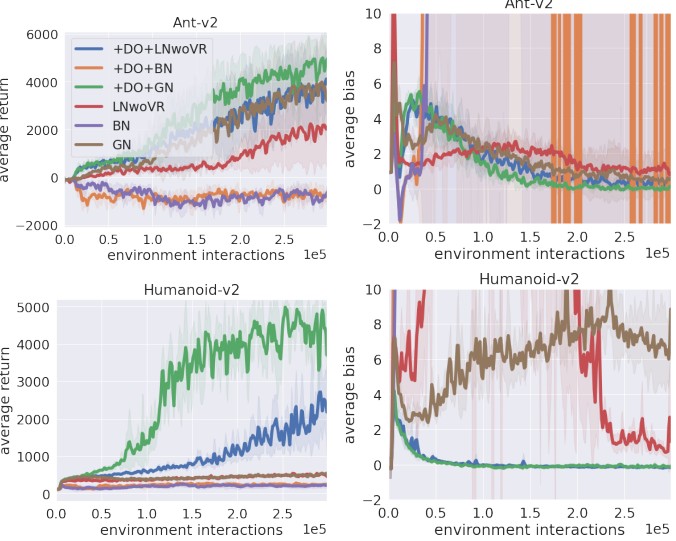

Figure 18: Average return and average estimation bias for six methods using batch normalization, group normalization, and the variant of layer normalization.

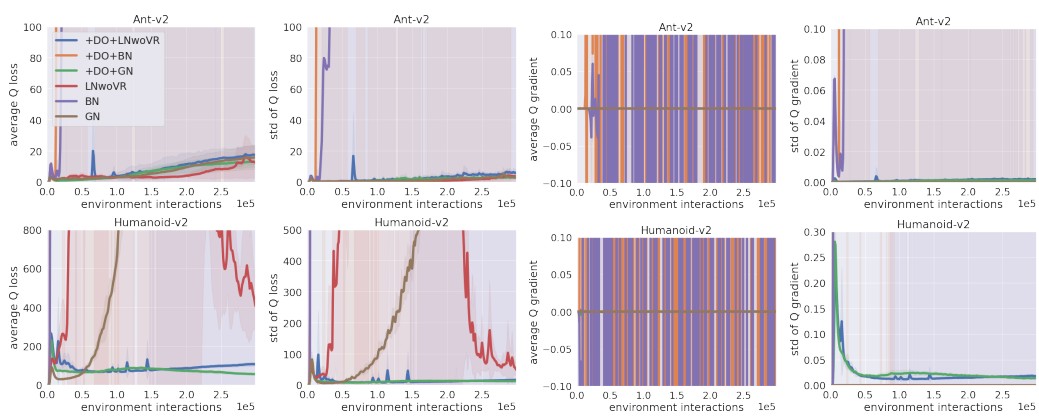

Figure 19: Average/variance of the Q-function loss (left part) and its gradient with respect to parameters (right part).

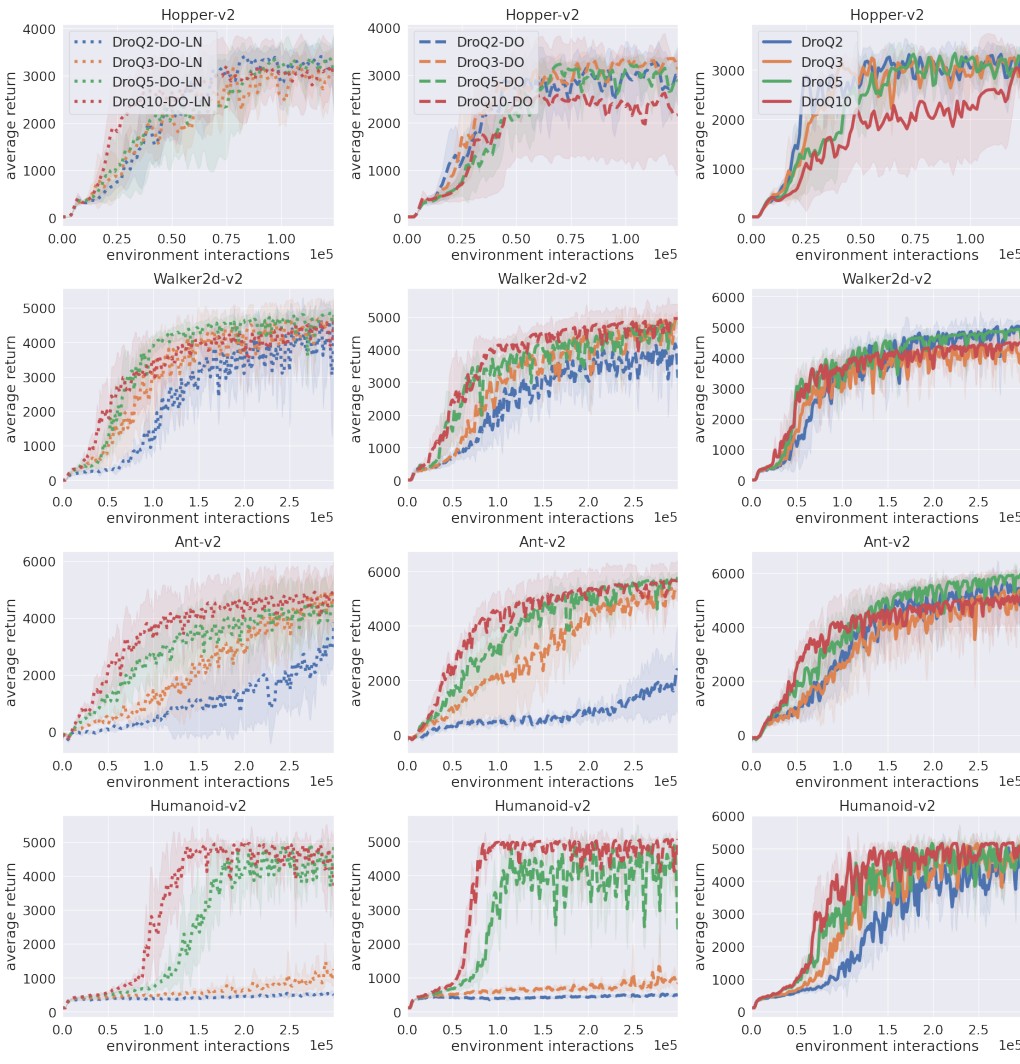

Figure 20: Additional ablation study result (average return). Left part (dotted lines): DroQ$N$-DO-LN, middle part (dashed lines): DroQ$N$-LN, right part (solid lines): DroQ$N$

## J  OUR SOURCE CODE

Our source code is available at `https://github.com/TakuyaHiraoka/Dropout-Q-Functions-for-Doubly-Efficient-Reinforcement-Learning`

