# OpenReview forum: "Dropout Q-Functions for Doubly Efficient Reinforcement Learning"
_ICLR.cc/2022/Conference — ICLR 2022 Poster_

### Official Review · Reviewer_1PuF · 2021-10-31

**Correctness:** 3
**Technical Novelty And Significance:** 2
**Empirical Novelty And Significance:** 3
**Recommendation:** 6
**Confidence:** 4

**Main Review:**

======Strengths=======

- Simplicity: the proposed method requires minimal modification to existing methods, making it easy to implement and use
- Significance: REDQ has achieved very strong sample efficiency but the ensemble of Q network leads to slower computation. How to achieve the same sample efficiency while reducing computation seems to be a very important question to solve.
- Analysis: comparison on computation and memory is good, shows significant reduction in computation and memory usage
- Results: results is nice, showing that the proposed method can achieve same sample efficiency of REDQ while being much more computation and memory efficient.

======Weaknesses=======

- Clarity of writing: some of the writing is not very accurate or not clear and can be refined
- Lack of in-depth investigation on why dropout and layernorm alone do not work well, and why dropout+layerrnorm works well, lacks comparison to other regularization techniques
- A number of other issues in the paper
- Proposed method lacks novelty: the proposed method (dropout+layernorm) seems to be a bit incremental.

Concerns, comments and questions:
- In page 1, "the UTD ratio is defined as the number of Q-function updates...", in the REDQ paper, this is actually defined as "the number of updates taken by the agent compared to the number of actual interactions with the environment", which is slightly different from your definition here (only the Q update), although for the main results reported in the REDQ paper, they focus on only changing the Q update number to isolate the effect of the policy update, you might want to be more clear as to prevent confusion. In your case, you can further clarify your writing, perhaps be more explicit and say its general definition and then say in your paper you define it to be only focused on Q update.
- Authors should check the writing again for small typos and mistakes. For example, should be MuJoCo instead of Mujoco. Figure 2 typo "and average and"
- top of page 2, "REDQ runs more than 2 times slower than SAC", you might want to further clarify here (and in other places in the paper where you mention this), are you talking about the computation time per update (where REDQ is a bit slower due to ensemble)? Or per environment interaction collected (where REDQ is a lot slower due to ensemble and high UTD)? Or the computation time for running until a performance level (where REDQ is not that slow due to reduced sample usage, I feel like this one is what you are talking about)? It is very important to make this point crystal clear since a majority of your discussion is on this subject. And helping the reader to gain a clear idea of exactly how much slower REDQ can be would help readers see the significance of your contribution.
- What is the full name of DUVN? Is this sth proposed in previous work? It is a bit unclear how this variant relates to your proposed method. Might want to explain it in a bit more detail.
- page 6 definition of the error (bias), the current definition is a little weird, it might be better to define the error as the difference of Q values, and then say we can estimate this error with MC return, instead of defining Q as the MC return. (I believe typically Q value is defined as expectation of MC return, not the other way around)
- end of page 6 "the most memory-intensive process... (applying the ReLU layer)" are you saying relu activation takes a lot of memory?
- Figure 2 hopper result seems to be lower than reported in the REDQ paper, not sure why?
- Ablation shows that layer norm helps with performance, but in the paper there is very limited discussion and analysis on why exactly layer norm makes dropout work better. What is so special about layer norm? Previously dropout had mixed results in DRL, why layer norm can fix the problems? What about other regularization schemes? Adding comparison to other schemes, and more analysis on how layer norm and dropout work together, either theoretically or empirically can help improve the significance of this paper. Otherwise the layernorm seems to be just a random hack added to boost performance.
- Though the name of Dr.Q is technically different from DrQ, I'm still a bit concerned whether this will lead to confusion


**Summary Of The Paper:**

Authors propose to use dropout and layer norm to make Randomized ensembled double Q-learning more computationally efficient. The proposed algorithm uses simple modifications to achieve comparable sample efficiency while being faster in computation. Authors also conduct empirical analysis on how dropout can be used effectively.

**Summary Of The Review:**

I think the paper has potential to make some good contribution, the authors already obtained some good results showing that Dr.Q can achieve the same performance as REDQ while greatly reduce computation and memory consumption, which is very nice. However, what is lack is some more in-depth investigation on why the proposed method works. Why dropout on its own does not work? And why layer norm will deal with the difficulties? What are the unique advantage of this combination over other methodologies? Very little discussion and empirical/theoretical analysis is provided in the paper. I think if the authors try explore these questions more, they should be able to make a bigger contribution and make this paper much stronger.

Though the paper studies an important problem, and has some good results, due to this major concern, and considering other issues in the paper, I don't quite think the paper is ready for publication in its current state, thus I currently lean towards rejection.

==============================================================

Post-rebuttal: after reading the authors' response and the revision, I feel that a large number of my concerns are addressed and the new results ablation and analysis look quite interesting. Though I still have some concerns, I now increase my score to 6.

---

> ### Author Response · Authors · 2021-11-22
> **Reply 1**
>
> Thank you very much for your valuable comments and suggestions.
> We have updated our paper based on your comments. Major changes we made are highlighted in blue.
>
> **Q1:**
> Clarity of writing: some of the writing is not very accurate or not clear and can be refined
> **Q1':**
> A number of other issues in the paper
> **A1:**
> We have revised our paper on the basis of your and the other reviewers' comments.
>
> **Q2:**
> Lack of in-depth investigation on why dropout and layernorm alone do not work well, and why dropout+layerrnorm works well, lacks comparison to other regularization techniques
> **Q2':**
> Ablation shows that layer norm helps with performance, but in the paper there is very limited discussion and analysis on why exactly layer norm makes dropout work better. What is so special about layer norm? Previously dropout had mixed results in DRL, why layer norm can fix the problems? What about other regularization schemes? Adding comparison to other schemes, and more analysis on how layer norm and dropout work together, either theoretically or empirically can help improve the significance of this paper. Otherwise the layernorm seems to be just a random hack added to boost performance.
> **Q2'':**
> However, what is lack is some more in-depth investigation on why the proposed method works. Why dropout on its own does not work? And why layer norm will deal with the difficulties?  What are the unique advantage of this combination over other methodologies? Very little discussion and empirical/theoretical analysis is provided in the paper. I think if the authors try explore these questions more, they should be able to make a bigger contribution and make this paper much stronger.
>
> **A2:**
> In this revision, we conducted more detailed analyses of why the combination of dropout and layer normalization works well (Appendices D, E, and F).
> We also analyzed the effect of normalization methods other than layer normalization (e.g., batch normalization) (Appendix E).
> Brief summary of these analyses is as follows:
>  - Appendix D: We analyzed how the mean and variance of Q-function learning loss and its gradient with respect to parameters change during the learning of Dr.Q and its variants. We found that dropout destabilizes Q-function learning (gradient variance), but layer normalization re-stabilizes it.
>  - Appendix E: We examined the effect of normalization methods (e.g., batch normalization) other than layer normalization. Its result is summarized as follows:
>    - Batch normalization did not improve performance well.
>    - Group normalization, which normalizes input by using the mean and variance over input elements as with layer normalization, improved performance well when combined with dropout.
>    - Layer normalization without variance rescaling did not perform well even when combined with Dropout. *This indicates that rescaling variance is one of the primal factors for producing synergy of dropout and layer normalization.*
>  - Appendix F: We examined the relation between (i) the synergy of dropout and layer normalization and (ii) the size of the Q-functions ensemble. Experimental results show that the synergy of dropout and layer normalization is high when the ensemble size is small. On the other hand, the effect of layer normalization itself is high when the ensemble size is large. This indicates that the magnitude of the synergy of dropout and layer normalization depends on the size of the ensemble.
>
>
> **Q3:**
> Proposed method lacks novelty: the proposed method (dropout+layernorm) seems to be a bit incremental.
>
> **A3:**
> As you (and Reviewer RMuR) pointed out, dropout and layer normalization have been individually presented in previous studies.
> However, we found new insights on how to combine them effectively (contribution 3 in the last paragraph of Section 1).
> In addition, in this revision, as we mentioned above, we conducted new additional analyses to explain why this combination works well (Appendices D, E, and F).
> The analyses have not been presented in previous studies, and we believe that they strengthen the novelty of our paper.
>
>
> **Q4:**
> In page 1, "the UTD ratio is defined as the number of Q-function updates...", in the REDQ paper, this is actually defined as "the number of updates taken by the agent compared to the number of actual interactions with the environment", which is slightly different from your definition here (only the Q update), although for the main results reported in the REDQ paper, they focus on only changing the Q update number to isolate the effect of the policy update, you might want to be more clear as to prevent confusion. In your case, you can further clarify your writing, perhaps be more explicit and say its general definition and then say in your paper you define it to be only focused on Q update.
>
> **A4:**
> We have made a revision to use the (general) definition of the UTD ratio presented in the REDQ paper

---

> ### Author Response · Authors · 2021-11-22
> **Reply 2**
>
> **Q5:**
> Authors should check the writing again for small typos and mistakes. For example, should be MuJoCo instead of Mujoco. Figure 2 typo "and average and"
>
> **A5:**
> We corrected typos and mistakes you pointed out, and also corrected other typos we found.
>
> **Q6:**
> top of page 2, "REDQ runs more than 2 times slower than SAC", you might want to further clarify here (and in other places in the paper where you mention this), are you talking about the computation time per update (where REDQ is a bit slower due to ensemble)? Or per environment interaction collected (where REDQ is a lot slower due to ensemble and high UTD)? Or the computation time for running until a performance level (where REDQ is not that slow due to reduced sample usage, I feel like this one is what you are talking about)? It is very important to make this point crystal clear since a majority of your discussion is on this subject. And helping the reader to gain a clear idea of exactly how much slower REDQ can be would help readers see the significance of your contribution.
>
> **A6:**
> We here mean the computation time per update. We revised the paper to make it clear.
>
>
> **Q7:**  What is the full name of DUVN? Is this sth proposed in previous work? It is a bit unclear how this variant relates to your proposed method. Might want to explain it in a bit more detail.
>
> **A7:**
> The full name of DUVN is Double Uncertain Value Network, which is proposed in [1].
>
> The original DUVN [7] injects model uncertainty into the target using a single dropout Q-function as $y = r + \gamma Q_{\text{Dr}, \bar{\phi}_1}(s', a')$.
>
> Here, $ Q_{\text{Dr}, \bar{\phi}_1}(s', a') $ is a dropout Q-function without layer normalization.
> Dr.Q variant (in which Dr.Q target is replaced with the above target + an entropy bonus) is the DUVN described in Section 4.1 in our paper.
> We have revised the paper to make these points clearer.
>
> [1] Thomas M Moerland, Joost Broekens, and Catholijn M Jonker. Efficient exploration with double uncertain value networks. arXiv preprint arXiv:1711.10789, 2017.
>
>
>
>
> **Q8:**
> page 6 definition of the error (bias), the current definition is a little weird, it might be better to define the error as the difference of Q values, and then say we can estimate this error with MC return, instead of defining Q as the MC return. (I believe typically Q value is defined as expectation of MC return, not the other way around)
>
> **A8:**
> We revised the paper as you suggested.
>
>
> **Q9:**
> end of page 6 "the most memory-intensive process... (applying the ReLU layer)" are you saying relu activation takes a lot of memory?
>
> **A9:**
> Yes. ReLU activation (in hidden layers) takes a lot of memory, and it is one of the most memory intensive parts.
> We have revised the paper to make this point clearer.
>
>
>
> **Q10:**
> Figure 2 hopper result seems to be lower than reported in the REDQ paper, not sure why?
>
> **A10:**
> This would be because the codebase of the methods used in our experiments (REDQ and SAC) is different from the one used in the REDQ paper.
> (We have used the codebase provided in the REDQ paper, but have failed to reproduce the results of the REDQ paper in almost all environments. So, we decided to use other well-maintained SAC codebase and implement REDQ and the other methods on top of it.)
> We have conducted many code reviews and preliminary experiments to replicate the result in Hopper, but it is still unclear which part of the codebase is responsible for it. (It is also still unclear why the score is lower only for the Hopper case.) (Perhaps, lower layers part such as Pytorch or Cuda is responsible for it.)
>
> Recently, the codebase provided in the REDQ paper was updated with major bug fixes (https://github.com/watchernyu/REDQ), so we will conduct additional experiments by using the latest version of this codebase.
>
>
>
> **Q11:**
> Though the name of Dr.Q is technically different from DrQ, I'm still a bit concerned whether this will lead to confusion
>
> **A11:**
> We are currently considering other better name.
>
> Again, thank you so much for your very beneficial review comments.
> If you have any further questions or suggestions, we would be more than happy to answer them in the next revision.

---

> > ### Comment · Reviewer_1PuF · 2021-11-29
> > **Response to rebuttal**
> >
> > I want to thank the authors for your effort in addressing my concerns and providing more analysis results.
> > For reproducing the results, indeed an earlier version of REDQ source code has an issue on terminal signal which might lead to performance issues. For Hopper, not sure what happened, perhaps it got stuck in a local optima? How many data do you use for initial exploration?
> >
> > I find your new results in appendix D, E, F to be very interesting and I feel now the paper offers some very interesting novel insight.
> >
> > For the next version of your paper, here are some suggestions:
> > - For appendix D perhaps expand a little on explanation of figure 9 to make things more clear for the reader. Currently it is a bit too concise.
> > - Figure 10 typo: bath normalization
> > - Appendix E maybe add a bit more discussion on how group norm is different from layer norm in your implementation.
> >
> > Though I still have some concerns, I do feel that a large number of my comments are addressed in the revision. So I will now increase my score to 6.

---

> > > ### Author Response · Authors · 2021-11-30
> > > **Reply**
> > >
> > > Thank you so much for your suggestions and feedback on our revision.
> > > We will reflect your suggestions in the next revision.
> > >
> > > **Q1:**   For Hopper, not sure what happened, perhaps it got stuck in a local optima? How many data do you use for initial exploration?
> > >
> > > **A1:**  As in the REDQ paper, the number of initial exploration steps is set to 5000 in Hopper.
> > > We will check if the policy is stuck in local optima, by varying the number of the initial exploration steps.

---

### Official Review · Reviewer_RMuR · 2021-11-03

**Correctness:** 3
**Technical Novelty And Significance:** 3
**Empirical Novelty And Significance:** 3
**Recommendation:** 6
**Confidence:** 3

**Main Review:**

Strengths:

This paper is well-written and easy to understand. The authors present a compelling reason to use ensemble-based Q-learning and why ensembles can be harder to work with from a computational perspective. I thinkt he obser The results and analysis support the hypothesis that dropout and layer normalization can have the same effect as a larger ensemble.


Weaknesses:

I believe novelty is a major weakness for this approach. Many of the individual components presented in this paper (dropout for Q functions, ensembles, etc) have been discussed in previous works. The authors claim that these have not been studied in high UTD settings, however the authors do not present any evidence that there is a difference between how REDQ and their method work at high UTD or lower UTD. Such a set of experiments will really help make this paper's claim stronger. Additionally, since the authors use M=2, this approach is very similar to a vanilla SAC implementation. It would be good to see an analysis of SAC with the dropout architecture (essentially this method but without any of the improvements from REDQ).

**Summary Of The Paper:**

This paper attempts to address computational efficiency in Ensemble-based Q-learning. The authors claim that previous, REDQ is too inefficient computationally as it requires too many Q-networsk in the ensemble. Thus, they propose an alternative approach that employs dropout (and layer normalization to stabilize the dropout layers), in order to get the same benefit of a larger ensemble in uncertainty estimation, while keeping computational costs low. The results and analysis on standard continuous control benchmark tasks show that the proposed approach maintains all the benefits of REDQ without the computational costs.

**Summary Of The Review:**

The paper presents a useful and interesting improvement to REDQ but novelty claims are not backed by experiments.

---

> ### Author Response · Authors · 2021-11-22
> **Reply 1**
>
> Thank you very much for your valuable comments and suggestions.
> We have updated our paper based on your comments. Major changes we made are highlighted in blue.
>
> **Q1:**
> Many of the individual components presented in this paper (dropout for Q functions, ensembles, etc) have been discussed in previous works. The authors claim that these have not been studied in high UTD settings, however the authors do not present any evidence that there is a difference between how REDQ and their method work at high UTD or lower UTD. Such a set of experiments will really help make this paper's claim stronger.
>
> **A1:**
> We have revised the paper to make it clearer that previous dropout/normalization methods for RL, which are discussed in previous works, do not work well in high UTD settings.
>
> Regarding the previous dropout methods for RL, we incorporated them into an RL framework and evaluated their performance in high UTD settings.
> - In Section 4.1, we evaluated the performance of DUVN, which is an RL framework using the dropout method proposed in [1, 2].
> - In Appendix A.3, we evaluated the performance of Sin-Dr.Q, which is an RL framework using the dropout method proposed in [3].
>
> In these sections, we showed that performances of DUVN and Sin-Dr.Q are significantly lower than those of REDQ and Dr.Q.
> We made a revision to make it clear that DUVN and Sin-Dr.Q are using dropout methods for RL proposed in previous works [1, 2, 3].
>
> Regarding previous normalization methods for RL, we evaluated the performance of using existing normalization methods (layer normalization and batch normalization) alone.
>  - In section 4.3, we showed that, in high UTD settings, an RL framework does not work well when it uses layer normalization alone as in [4, 5, 6].  Comparing the results shown in Figures 2 and 3, we can see that the framework using layer normalization alone (-LN) does not work as well as REDQ. We revised the paper to make this point clear.
>  - In this revision, we conducted a new evaluation for an RL framework that uses batch normalization in high UTD setting (Appendix E). The evaluation result shows that the framework doe not work well in the high UTD. (Using batch normalization in low UTD settings have already been discussed in previous research).
>
> [1] Cosmo Harrigan. Deep reinforcement learning with regularized convolutional neural fitted Q iteration.
> [2] Thomas M Moerland, Joost Broekens, and Catholijn M Jonker. Efficient exploration with double uncertain value networks. arXiv preprint arXiv:1711.10789, 2017.
> [3] Natasha Jaques, Asma Ghandeharioun, Judy Hanwen Shen, Craig Ferguson, Agata Lapedriza, Noah Jones, Shixiang Gu, and Rosalind Picard. Way off-policy batch deep reinforcement learning of implicit human preferences in dialog. arXiv preprint arXiv:1907.00456, 2019.
> [4] Matt Hoffman, Bobak Shahriari, John Aslanides, Gabriel Barth-Maron, Feryal Behbahani, Tamara Norman, Abbas Abdolmaleki, Albin Cassirer, Fan Yang, Kate Baumli, Sarah Henderson, Alex Novikov, Sergio Gomez Colmenarejo, Serkan Cabi, Caglar Gulcehre, Tom Le Paine, Andrew Cowie, ZiyuWang, Bilal Piot, and Nando de Freitas. Acme: A research framework for distributed reinforcement learning. arXiv preprint arXiv:2006.00979, 2020.
> [5] Xiaoteng Ma, Li Xia, Zhengyuan Zhou, Jun Yang, and Qianchuan Zhao. DSAC: Distributional soft actor critic for risk-sensitive reinforcement learning. In Reinforcement Learning for Real Life Workshop at ICML 2019, 2020.
> [6] Amy Zhang, Rowan Thomas McAllister, Roberto Calandra, Yarin Gal, and Sergey Levine. Learning invariant representations for reinforcement learning without reconstruction. In Proc. ICLR, 2021.
>
>
> **Q2:**
> Additionally, since the authors use M=2, this approach is very similar to a vanilla SAC implementation. It would be good to see an analysis of SAC with the dropout architecture (essentially this method but without any of the improvements from REDQ).
>
> **A2:**
> We added performance evaluation results for SAC variants that use the dropout Q-function (Figure 13 in Appendix G).
> The results show that there is no significant difference between Dr.Q and the SAC variant with the dropout Q-function (dropout+layer normalization).
> The results also show that performances of the SAC variant that uses layer normalization alone and variant that uses dropout alone were basically the same as those of Dr.Q variants (using either dropout or layer normalization) shown in Figure 3 in Section 4.3.

---

> > ### Comment · Reviewer_RMuR · 2021-11-29
> > **Response to Authors**
> >
> > Dear Authors,
> >
> > Thank you for taking my suggestions into consideration. I think the empirical analysis added does help and I am increasing my rating.
> >
> > Best regards

---

> > > ### Author Response · Authors · 2021-11-30
> > > **Reply**
> > >
> > > Thank you so much for your feedback on our response.

---

> ### Author Response · Authors · 2021-11-22
> **Reply 2**
>
> **Q3:**
> I believe novelty is a major weakness for this approach. Many of the individual components presented in this paper (dropout for Q functions, ensembles, etc) have been discussed in previous works.
>
> **A3:**
> As you pointed out, the individual components themselves have been presented in previous studies.
> However, we found new insights on how to combine them effectively (contribution 3 in the last paragraph of Section 1).
> In addition, in this revision, we conducted new additional analyses to explain why this combination works well (Appendices D, E, and F).
> These analyses have not been presented in previous studies, and we believe that they strengthen the novelty of our paper.
> We also believe that investigating effective combinations of known components (e.g., as in [7]) is as beneficial to the RL community as proposing brand new components.
>
> [7] Hessel, Matteo, et al. "Rainbow: Combining improvements in deep reinforcement learning." Thirty-second AAAI conference on artificial intelligence. 2018.
>
> If you have any further questions or suggestions, we would be more than happy to answer them in the next revision.

---

### Official Review · Reviewer_fiFc · 2021-11-04

**Correctness:** 4
**Technical Novelty And Significance:** 3
**Empirical Novelty And Significance:** 3
**Recommendation:** 6
**Confidence:** 4

**Main Review:**

Strengths:

This was a very interesting paper. Contrary to the ensemble approach for bias reduction which is rapidly gaining popularity in recent years, the authors proposed an alternative which allows an SAC-like algorithm to work in high UTD settings by adding dropout and layernorm, which in my opinion is the main contribution of this paper. The presentation of the paper is quite clear and it was very easy to read. Experiments are quite thorough and very well done. The ablation studies are also quite clear in presenting how the different components of the algorithm affect the performance of the algorithm.

Weakness:

My main concern is that it is not entirely clear from the paper why the method presented by the authors work so well. Dropout, layernorm, and high UTD have all be separately applied to and studied in off-policy algorithms and it is not clear why combining all three leads to such a boost in performance. Intuitively dropout has similar effects as ensemble models since they add noise to the Q networks thereby injecting uncertainty into the target, but based on the authors' ablation studies and previous work, having dropout alone is not sufficient. More analysis on this front would do much to strengthen the authors' overall arguments.

One final minor point, I suggest modifying the name Dr. Q, I don't think adding the period after Dr will help with avoiding name conflicts with DrQ as the authors have suggested.

**Summary Of The Paper:**

This work proposes a new algorithm based on REDQ. The new algorithm Dr. Q. uses a small ensemble of dropout Q functions along with layer normalization. The proposed algorithm achieves similar sample efficiency as REDQ but better computational efficiency.

**Summary Of The Review:**

Though the paper could benefit from additional analyze on explaining the performance of the proposed algorithm, I think this paper brings about some valuable insights which may benefit and impact how off-policy algorithms are implemented in the long-run.

---

> ### Author Response · Authors · 2021-11-22
> **Reply**
>
> Thank you very much for your valuable comments and suggestions.
> We have updated our paper based on your comments. Major changes we made are highlighted in blue.
>
> **Q1:**
> My main concern is that it is not entirely clear from the paper why the method presented by the authors work so well. Dropout, layernorm, and high UTD have all been separately applied to and studied in off-policy algorithms and it is not clear why combining all three leads to such a boost in performance. Intuitively dropout has similar effects as ensemble models since they add noise to the Q networks thereby injecting uncertainty into the target, but based on the authors' ablation studies and previous work, having dropout alone is not sufficient. More analysis on this front would do much to strengthen the authors' overall arguments.
>
> **A1:**
> Thank you for the great suggestion. We have revised the paper by adding new analyses to answer the following questions:
>  - Why using dropout alone is not sufficient in high UTD?
>  - Why does the combination of dropout and layer normalization work well?
>
> The new analyses are added as Appendices D, E, and F in the revised paper. In sum,
>   - Appendix D: We analyzed how the mean and variance of Q-function learning loss and its gradient with respect to parameters change during the learning of Dr.Q and its variants. We found that dropout destabilizes Q-function learning (gradient variance), but layer normalization re-stabilizes it.
>   - Appendix E: We examined the effect of normalization methods (e.g., batch normalization) other than layer normalization. Its result is summarized as follows:
>     - Batch normalization did not improve performance well.
>     - Group normalization, which normalizes input by using the mean and variance over input elements as with layer normalization, improved performance well when combined with dropout.
>     - Layer normalization without variance rescaling did not perform well even when combined with Dropout. *This indicates that the rescaling of the variance is one of the primal factors for producing synergy of dropout and layer normalization.*
>   - Appendix F: We examined the relation between (i) the synergy of dropout and layer normalization and (ii) the size of the Q-functions ensemble. Experimental results show that the synergy of dropout and layer normalization is high when the ensemble size is small. On the other hand, the effect of layer normalization itself is high when the ensemble size is large. This indicates that the magnitude of the synergy of dropout and layer normalization depends on the size of the ensemble.
>
>
> **Q2:**
> One final minor point, I suggest modifying the name Dr. Q, I don't think adding the period after Dr will help with avoiding name conflicts with DrQ as the authors have suggested.
>
> **A2:**
> We are currently considering other better name.
>
> If you have any further questions or suggestions, we would be more than happy to answer them in the next revision.

---

> > ### Comment · Reviewer_fiFc · 2021-11-29
> > **Response to revision**
> >
> > I would like to thank the authors for their response and the additional experiments+analysis in their revision. I agree with reviewer 1PuF in that the new sections the authors added to the appendix offer quite a bit of additional insight to their existing results, especially how layernorm helps to mitigate the instability caused by dropout. Overall I am satisfied with how the authors addressed my concerns.

---

> > > ### Author Response · Authors · 2021-11-30
> > > **Reply**
> > >
> > > Thank you so much for your feedback on our revision.

---

### Public Comment · ~Qiang_He1 · 2021-11-21
**Missing Potential Related Work**

The proposed method looks very interesting, while the discussion of a potential related work,  MEPG[1], is missing. MEPG introduces minimalist ensemble consistent Bellman update, which is, to some extend, relevant to Dr.Q.

[1] He, Qiang, et al. "MEPG: A Minimalist Ensemble Policy Gradient Framework for Deep Reinforcement Learning." arXiv preprint arXiv:2109.10552 (2021).

---

> ### Author Response · Authors · 2021-11-22
> **Reply**
>
> **Q1:**
> The proposed method looks very interesting, while the discussion of a potential related work, MEPG[1], is missing. MEPG introduces minimalist ensemble consistent Bellman update, which is, to some extend, relevant to Dr.Q.
> [1] He, Qiang, et al. "MEPG: A Minimalist Ensemble Policy Gradient Framework for Deep Reinforcement Learning." arXiv preprint arXiv:2109.10552 (2021).
>
> **A1:**
> Thank you so much for your reference.
> We added a discussion about the related work [1] to the revised paper.
>
> Guessing that you are the author of [1], we think the following points in our revised paper are interesting for you:
>  - The comparison results between Dr.Q and DUVN (Figure 2 in Section 4.1) and those between Dr.Q with Sin-Dr.Q (Figure 6 in Appendix A.3) show that it is important to "use multiple dropout Q-functions" in high UTD settings, which is in contrast to your result "using a single dropout Q-function is sufficient" in low UTD settings. In addition, our analyses (Sections 4.3, D, and E) show that layer normalization is also important for improving dropout Q-functions in high UTD settings. Further, our analyses in Appendix F show how the effect of dropout (and layer normalization) changes as the size of the Q-function ensemble changes in high UTD settings. These findings could be useful when you conduct some experiments in high UTD settings.
>  - In Appendix C, we show that introducing dropout to both current and target Q-functions improves performance in high UTD settings. This result would demonstrate the usefulness of your claim "Minimalist ensemble consistent Bellman update" (i.e. "introduce dropout in both current and target Q-functions") in high UTD settings.

---

### Public Comment · ~Takuya_Hiraoka1 · 2022-03-14
**Updates on our camera-ready**

We made the following updates:
1. Fix typos and modify unclear explanations/discussions.
2. Rename our proposed method as Dr.Q -> DroQ.
3. Add results of reproducing our experiments on the REDQ codebase (Appendix I).
4. Add a link to our source code (Appendix J).
5. Add a poster and slides to the supplementary material.

---

### Decision · Program_Chairs · 2022-01-20

**Decision:**

Accept (Poster)

**Comment:**

The manuscript describes a method for improving the computational efficiency of randomized ensemble double Q-learning for continuous action RL, by using a small ensemble of Q-functions equipped with dropout and layer normalization, achieving matched sample efficiency at considerably less computational cost.

Reviewers praised the method's simplicity and achievement of its stated objective of reducing the computational cost of deploying ensemble Q functions. In general, the paper was found to be easy to understand and well written. Several expressed concern about the lack of interrogation of why this combination of dropout and layer norm worked so well and an overall lack of novelty. Other miscellaneous criticisms were well-addressed in rebuttal and extensive new analyses in the Appendix were noted by several reviewers as adding much to the work.

In the AC's opinion, this is an example of a simple but non-obvious combination of well-known ideas that works very well. The review process has improved the level of empirical rigor that has gone into understanding the properties and trade-offs of this method. I'm happy  to recommend acceptance, though would echo reviewers concerns that dubbing the method "Dr.Q" will lead to confusion and would strongly urge adopting another name for the camera ready.